# *WizardLM*: Empowering Large Pre-Trained Language Models to Follow Complex Instructions

**Can Xu**[1*]   **Qingfeng Sun**[1*]   **Kai Zheng**[1*]   **Xiubo Geng**[1]   **Pu Zhao**[1]
**Jiazhan Feng**[2†]   **Chongyang Tao**[1]   **Qingwei Lin**[1]   **Daxin Jiang**[1‡]
[1]Microsoft
[2]Peking University
{caxu,qins,zhengkai,xigeng,puzhao,chongyang.tao,qlin,djiang}@microsoft.com
{fengjiazhan}@pku.edu.cn

## ABSTRACT

Training large language models (LLMs) with open-domain instruction following data brings colossal success. However, manually creating such instruction data is very time-consuming and labor-intensive. Moreover, humans may struggle to produce high-complexity instructions. In this paper, we show an avenue for creating large amounts of instruction data with varying levels of complexity using LLM instead of humans. Starting with an initial set of instructions, we use our proposed *Evol-Instruct* to rewrite them step by step into more complex instructions. Then, we mix all generated instruction data to fine-tune LLaMA. We call the resulting model *WizardLM*. Both automatic and human evaluations consistently indicate that *WizardLM* outperforms baselines such as Alpaca (trained from Self-Instruct) and Vicuna (trained from human-created instructions). The experimental results demonstrate that the quality of instruction-following dataset crafted by *Evol-Instruct* can significantly improve the performance of LLMs.

## 1    INTRODUCTION

Large-scale language models (LLMs) have become the go-to approach for numerous natural language processing tasks (Brown et al., 2020; Ouyang et al., 2022; Touvron et al., 2023). LLMs are trained on large volumes of text data to predict the subsequent tokens, enabling them to generate coherent and fluent text in response to various inputs. However, these models often struggle to follow instructions or goals specified by users, which limits their usefulness and applicability in real-world scenarios.

The NLP community has recently witnessed many endeavors to train LLMs to follow instructions better and be more helpful  (Zhao et al., 2023; He et al., 2023; Guo et al., 2023; Li et al., 2023b). Initial attempts (Aribandi et al., 2022; Wei et al., 2021; Xu et al., 2022; Sanh et al., 2022; Chung et al., 2022) to train instruction-following language models are based on a collection of various NLP tasks, with a small amount of hand-written instructions. These closed-domain instructions suffer from two main drawbacks: first, all the samples in an NLP dataset share only a few common instructions, severely limiting their diversity; second, the instructions usually only ask for one task. But in real life, human instructions often have multiple and varied task demands. By using open-domain instruction data generated by real human users, OpenAI's LLMs (e.g., InstructGPT (Ouyang et al., 2022) and ChatGPT [1]) have achieved great success. These open-domain instructions can fully unleash the unlimited potential of LLMs (Luo et al., 2023; Ma et al., 2023; Hu et al., 2023; Zhu et al., 2023) and enable them to perform more complex and diverse tasks. However, using humans to create open-domain instruction datasets like OpenAI did will encounter the following challenges. The whole annotating process is extremely expensive and time-consuming (Kopf et al., 2023; Chen et al., 2023; Sun et al., 2023; Yuan et al., 2023). On the other hand, the difficulty level distribution of human-created instructions is skewed towards being easy or moderate, with fewer difficult ones (according

---

*    Equal contribution.
†    Work done during the internship at Microsoft.
‡    Corresponding author.
[1]https://chat.openai.com/

to the difficulty statistics of ShareGPT (Chiang et al., 2023) from Figure 5a). Human annotators are prone to fatigue and cannot sustain high-intensity work to produce a sufficient proportion of high-difficulty instructions (Zhang et al., 2023; Xiao et al., 2023; Manakul et al., 2023; Zhong et al., 2023). Based on these issues, developing an automatic method that can mass-produce open-domain instructions (especially the more difficult ones) at a relatively low cost becomes the key to further advancing instruction-tuned language models (Bao et al., 2023; Liu et al., 2023; Bian et al., 2023; Cabannes et al., 2023).

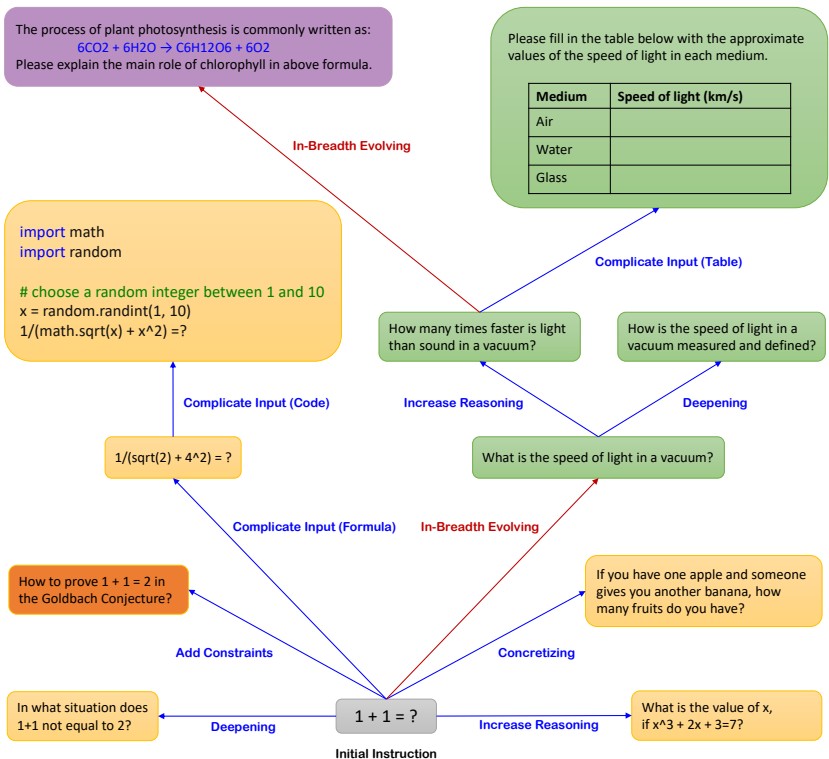

Figure 1: Running Examples of *Evol-Instruct*.

In this work, we introduce *Evol-Instruct*, a novel method using LLMs instead of humans to automatically mass-produce open-domain instructions of various difficulty levels, to improve the performance of LLMs. Figure 1 shows the running examples of *Evol-Instruct*. Starting from a simple initial instruction "1+1=?", our method randomly selects In-depth Evolving (blue direction line) or In-breadth Evolving (red direction line) to upgrade the simple instruction to a more complex one or create a new one (to increase diversity). The In-depth Evolving includes five types of operations: add constraints, deepening, concretizing, increase reasoning steps, and complicate input. The In-breadth Evolving is mutation, i.e., generating a completely new instruction based on the given instruction. These six operations are implemented by prompting an LLM with specific prompts. Since the evolved instructions are generated from LLMs, sometimes the evolving will fail. We adopt an instruction eliminator to filter the failed instructions, which is called Elimination Evolving. We repeat this evolutionary process for several rounds to obtain enough instruction data containing various complexities.

In order to verify the effectiveness of *Evol-Instruct* and whether the instructions it creates for fine-tuning surpass those created by humans, we evolve the instructions from Aplaca (Taori et al., 2023) data (created by machine), fine-tune the LLaMA (Touvron et al., 2023) model, and comprehensively compare the fine-tuned model *WizardLM* with Vicuna (Chiang et al., 2023) trained on ShareGPT (instructions are created by human). Alpaca data has a total of $52k$ samples and is generated using self-instruct (Wang et al., 2022a) from only $175$ human-created seed instructions. We choose Alpaca data as the initial data for evolution, which can ensure that the training instructions of *WizardLM* have almost no direct human participation in annotations. We execute four epochs of evolution

using OpenAI ChatGPT API[2] and finally obtain $250k$ instructions. To ensure a fair comparison with Vicuna's $70k$ real user data, we sampled $70k$ from the full $250k$ data and fine-tuned the LLaMA 13B model. Because the original Alpaca data only has $52k$ samples, we used its self-instruct method to generate an additional $18k$ data, and retrained the LLaMA 13B model with its code[3] to get Alpaca 13B as our baseline. Due to the low proportion of difficult instructions in the previous instruction-following test dataset, we manually created a new difficulty-balanced test dataset, named *WizardEval*. We evaluate Alpaca, Vicuna, ChatGPT, and WizardLM on a wide range of LLM benchmarks (covering reasoning, code, mathematics, general conversation, etc.). Our main findings are as follows:

- We introduce *Evol-Instruct*, a novel approach that enhances the performance of the open-source LLMs by a large margin via automatically mass-producing open-domain instructions of various topics and difficulty levels.

- We develop *WizardLM* model, which significantly surpasses typical open-source LLMs such as Alpaca and Vicuna in a series of benchmarks. Notably, *WizardLM* outperforms baselines by a substantial margin in terms of code, math, GPT-4 and human evaluations.

- We have undertaken a preliminary investigation that underscores the importance of instruction complexity in attaining outstanding performance in supervised fine-tuning large pre-trained language models.

## 2 RELATED WORK

**Closed domain instruction tuning**   Early instruction-following training work (Wei et al., 2021; Longpre et al., 2023) concerns cross task generalization in LMs, where LMs are fine-tuned on a broad range of public NLP datasets and evaluated on a different set of NLP tasks. T5 Raffel et al. (2020) made the earliest attempt by training natural language processing (NLP) tasks such as question answering, document summarization, and sentiment classification together using a unified text-to-text format. Works such as FLAN Wei et al. (2021), ExT5 Aribandi et al. (2022), T0 Sanh et al. (2022), and KnowDA Wang et al. (2022c) increased the number of NLP tasks to around one hundred, with several instructions carefully designed for each task de Wynter et al. (2023); Svikhnushina & Pu (2023); Huang et al. (2023); Yue et al. (2023). Furthermore, works such as ZeroPrompt Xu et al. (2022) and FLAN-T5 Chung et al. (2022) raised the number of tasks to the thousands. These studies consistently show that fine-tuning LMs with diverse NLP task instructions enhances their performance on new tasks. However, LLMs trained with these closed-form instructions (i.e., instructions are often only for a single NLP task, and the input data form is simple) tend to fail in real-world user scenarios.

**Open domain instruction tuning**   Our work belongs to this research line. OpenAI has hired many annotators and written many instructions with corresponding correct responses. These human-created instructions have diverse forms and rich task types. Based on this dataset, OpenAI trained GPT-3 Brown et al. (2020) into InstructGPT Ouyang et al. (2022), which can process a variety of real user instructions and led to the success of ChatGPT. Orca Mukherjee et al. (2023) learns not only the superficial response text from LLMs, but also captures complex reasoning process signals. Since these outstanding works from OpenAI were not open-sourced, Alpaca Taori et al. (2023) and Vicuna Chiang et al. (2023) subsequently actively explored open-domain instruction fine-tuning based on the open-source LLM LLaMA Touvron et al. (2023). Alpaca used a dataset of 50k instructions generated from a limited (e.g., 175 samples) seed set of manually-written instructions. Our work is different from InstructGPT and Vicuna in that we use AI-generated data for instruction fine-tuning. Unlike Alpaca's self-instruct Wang et al. (2022a) generation method, *Evol-Instruct* can control the difficulty and complexity level of the generated instructions.

## 3 APPROACH

In this section, we elaborate on the details of the proposed *Evol-Instruct*. As illustrated in Figure 2, the pipeline mainly contains two components: Instruction Evolver and Instruction Eliminator. The

---

[2] gpt-3.5-turbo from `https://oai.azure.com/portal`
[3] `https://github.com/tatsu-lab/stanford_alpaca`

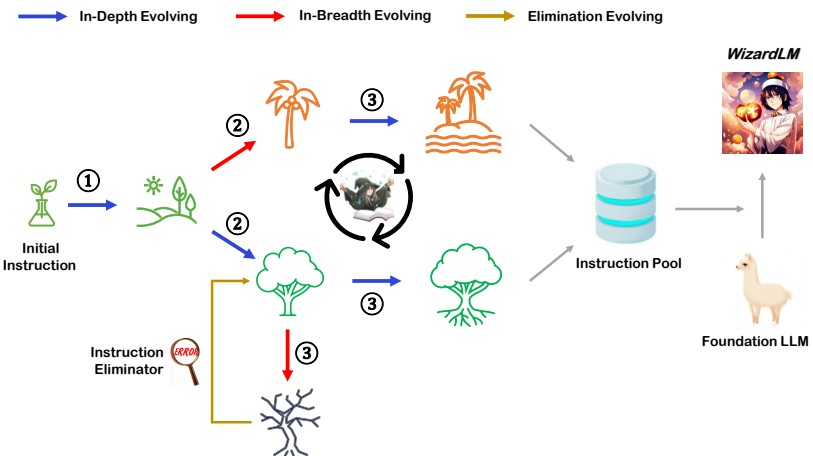

Figure 2: Overview of *Evol-Instruct*

details of these compoents will be presented in Sec. 3.2 and instruction fine-tuning method will be described in Sec. 3.3.

### 3.1 DEFINITION OF INSTRUCTION DATA EVOLUTION

We start the evolution from a given initial instruction dataset $D^{(0)} = (I_k^{(0)}, R_k^{(0)})_{1 \leq k \leq N}$, where $I_k^{(0)}$ is the $k$-th instruction in $D^{(0)}$, $R_k^{(0)}$ is the corresponding response for the $k$-th instruction, and $N$ is the number of samples in $D^{(0)}$. In each evolution, we upgrade all the $I^{(t)}$ in $D^{(t)}$ to $I^{(t+1)}$ by prompting a LLM with *Evol-Instruct* prompt, and then use the LLM to generate corresponding responses $R^{t+1}$ for the newly evolved $I^{t+1}$. Thus, we obtain an evolved instruction dataset $D^{t+1}$. By iteratively performing $M$ evolutions, we can sequentially obtain $M$ evolution datasets $[D^{(1)} \cdots D^{(M)}]$. Our work focuses on open-domain instruction data, where instructions have varying inputs and tasks without a clear distinction between the instruction part and the input.

### 3.2 AUTOMATIC INSTRUCTION DATA EVOLUTION

Our pipeline for instruction evolution consists of three steps: 1) instruction evolving, 2) response generation, and 3) elimination evolving, i.e., filtering intructions that fails to evolve.

**Instruction Evolution.** We found that LLMs can make given instructions more complex and difficult using specific prompts. Additionally, they can generate entirely new instructions that are equally complex but completely different. Using this discovery, we can iteratively evolve an initial instruction dataset, improving difficulty level and expanding its richness and diversity. We initiate the instruction pool with the given initial instruction dataset $D^{(0)}$. In each evolution epoch, upgraded instructions from the previous epoch are taken out from the pool. Then we leverage the instruction evolver to evolve each fetched instruction, and the instruction eliminator to check whether the evolution fails. Successful evolved instructions are added to the pool, while unsuccessful ones are placed back as they are, with the hope of upgrading them successfully in the next evolution epoch.

**Instruction Evolver.** The Instruction Evolver is an LLM that uses *Evol-Instruct* prompts to evolve instructions, with two types: in-depth evolving and in-breadth evolving.

In-Depth Evolving enhances instructions by making them more complex and difficult through five types of prompts: add constraints, deepening, concretizing, increased reasoning steps, and complicating input. The core part of In-Depth Evolving's prompt is "Your objective is to rewrite a given prompt into a more complex version to make those famous AI systems (e.g., ChatGPT and GPT4 (OpenAI, 2023)) a bit harder to handle. But the rewritten prompt must be reasonable, understood, and responded to by humans". We require the LLM to create challenging instructions that are reasonable and not arbitrarily imagined by AI. A gradual difficulty increase is necessary to avoid

filling the instruction set with extremely complex instructions, which would harm the generalization performance of trained models. To control difficulty increase, we make each evolution "a bit harder" and restrict adding a maximum of 10 to 20 words. Among the five mentioned evolving, all can be implemented without any in-context examples except for complicating input. We show the prompt of add constraints as follows (the prompts of deepening, concretizing and increased reasoning steps will be detailed in the Appendix A-C).

---

**Example 3.1: Prompt for Adding Constraints of In-Depth Evolving**

I want you act as a Prompt Rewriter.
Your objective is to rewrite a given prompt into a more complex version to make those famous AI systems (e.g., ChatGPT and GPT4) a bit harder to handle. But the rewritten prompt must be reasonable and must be understood and responded by humans.
Your rewriting cannot omit the non-text parts such as the table and code in #Given Prompt#. Also, please do not omit the input in #Given Prompt#.
You SHOULD complicate the given prompt using the following method:
**Please add one more constraints/requirements into #Given Prompt#**

You should try your best not to make the #Rewritten Prompt# become verbose, #Rewritten Prompt# can only add 10 to 20 words into #Given Prompt#. '#Given Prompt#', '#Rewritten Prompt#', 'given prompt' and 'rewritten prompt' are not allowed to appear in #Rewritten Prompt#

**#Given Prompt#:**
{Here is instruction.}
**#Rewritten Prompt#:**

---

For complicating input, we will use in-context demonstration. Due to the lengthy demonstrations, we will provide a brief template below, with the full prompt detailed in the Appendix D.

---

**Example 3.2: Prompt for Complicating Input of In-Depth Evolving**

I want you act as a Prompt Rewriter.

Your objective is to rewrite a given prompt into a more complex version to make those famous AI systems (e.g., ChatGPT and GPT4) a bit harder to handle. But the rewritten prompt must be reasonable and must be understood and responded by humans.

You must add **[XML data]** format data as input data in [Rewritten Prompt]
**#Given Prompt#:**
{Here is instruction of Example 1.}
**#Rewritten Prompt#:**
{Here is rewritten instruction of Example 1.}

**... N -1 Examples ...**

You must add **[#Given Dataformat#]** format data as input data in [Rewritten Prompt]
**#Given Prompt#:**
{Here is instruction of Example N.}
**#Rewritten Prompt#:**

---

In-Breadth Evolving aims to enhance topic coverage, skill coverage, and overall dataset diversity. Open-domain instruction finetune datasets (e.g., Alpaca, ShareGPT, etc.) are typically small in scale, lacking topic and skill diversity. To solve this problem, we designed a prompt to generate a completely new instruction based on the given instruction, requiring the new instruction to be more long-tailed. Our In-Breadth Evolving prompt is as follows:

---

**Example 3.3: Prompt for In-Breadth Evolving**

I want you act as a Prompt Creator.
Your goal is to draw inspiration from the #Given Prompt# to create a brand new prompt.
This new prompt should belong to the same domain as the #Given Prompt# but be even more rare.
The LENGTH and difficulty level of the #Created Prompt# should be similar to that of the #Given Prompt#.
The #Created Prompt# must be reasonable and must be understood and responded by humans.

---

> '#Given Prompt#', '#Created Prompt#', 'given prompt' and 'created prompt' are not allowed to appear in #Created Prompt#.
>
> **#Given Prompt#:**
> {Here is instruction.}
> **#Created Prompt#:**

**Response Generation.** We use the same LLM as for evolving to generate the corresponding responses for the evolved instructions. The generation prompt is "{Here is instruction.}", we feed it into the request of the ChatGPT-3.5 and parse the returned text body as the response.

**Elimination Evolving.** We classify the following four situations as instruction evolution failure:

1. The evolved instruction does not provide any information gain compared to the original one. We use ChatGPT to make this determination, details please refer to Appendix G.

2. The evolved instruction makes it difficult for the LLM to generate a response. We found that when the generated response contains "sorry" and is relatively short in length (i.e., less than 80 words), it often indicates that the LLM struggles to respond to the evolved instruction. So we can use this rule to make a judgment.

3. The response generated by the LLM only contains punctuation and stop words.

4. The evolved instruction obviously copies some words from the evolving prompt, such as "given prompt", "rewritten prompt", "#Rewritten Prompt#", etc.

### 3.3 FINETUNING THE LLM ON THE EVOLVED INSTRUCTIONS

Once all evolutions are done, we will merge the initial instruction dataset with evolved instruction data from all epochs and randomly shuffle the samples to create the fine-tuning dataset. This processing ensures even distribution of instructions of varying difficulty levels in the dataset, maximizing model fine-tuning smoothness. To prove that the performance gain is not due to the increased amount of data after merging, but from our proposed novel method *Evol-Instruct*, we randomly sample an equal amount of data the same with training baselines (e.g., Vicuna) from this merged data as our final fine-tuning data. We choose Vicuna's prompt as the prompt for our fine-tuning, the specific format is "A chat between a curious user and an artificial intelligence assistant. The assistant gives helpful, detailed, and polite answers to the user's questions. USER: Hi ASSISTANT: Hello. USER: Who are you? ASSISTANT: I am WizardLM ......."

## 4 EXPERIMENT

We assess *WizardLM*, Alpaca, Vicuna, and ChatGPT using both automatic and human evaluations.

### 4.1 BASELINES

(1) **ChatGPT** is an AI chatbot developed by OpenAI that can interact with users in a natural and engaging way. It is built on top of LLMs like GPT-3.5 and GPT-4, trained on vast internet text data.

(2) **Alpaca** is an open-source instruction-following model developed by Stanford University. For a fair comparison, we expanded the number of instructions from $52k$ to $70k$ using self-Instruct adopted by Alpaca and replaced the original Davici-003 responses with ChatGPT's responses. We re-trained Alpaca 13B from LLaMA 13B(Touvron et al., 2023) based on this new Alpaca data.

(3) **Vicuna** is based on LLaMA and fine-tuned on $70k$ user-shared conversations collected from ShareGPT. It is one of the most advanced and versatile open instruction-following models available today. We use the 13B-v1.1 model from FastChat [4].

(4) Open-source models trained from Llama 13B, including **Baize** (Xu et al., 2023), **CAMEL** (Li et al., 2023a), and **Tulu** (Wang et al., 2023)

---

[4] https://github.com/lm-sys/FastChat

## 4.2 EXPERIMENT DETAIL

To construct the dataset, we initialize it with the $52k$ instruction dataset of Alpaca and iteratively perform $M$ evolutions, where $M = 4$. For each instruction in each round of evolution, we randomly select one evolving prompt from total six prompts (i.e., five from in-depth evolving and one from in-breadth evolving) with equal probability. We execute above process using Azure OpenAI ChatGPT API[5]. Then, we leverage ChatGPT to generate responses. Finally, we obtain $250k$ instructions. For a fair comparison, we randomly sample $70k$ data from $250k$ data with equal probability as the final training data for *WizardLM*, the same as the amount of training data for Vicuna. We use a temperature of 1 to generate response and set the maximum number of tokens for generation to 2048. Additionally, we set the frequency penalty to zero and top-p to 0.9. Totally, we request the API $52k$ $\times 4 \times 3 = 624k$ times to construct the full dataset. We use pre-trained LLaMA 13B (Touvron et al., 2023) to initialize our model. We adopt Adam optimizer with an initial learning rate of $2 \times 10^{-5}$, a maximum number of tokens 2048, and the batch size is 4 for each GPU. We train our model on 8 V100 GPUs with Deepspeed Zero-3 for 140 hours on 3 epochs. For inference, we use greedy search for *WizardLM* and baseline models, and set the maximum generation length to 2048.

## 4.3 AUTOMATIC EVALUATION

To present a comprehensive overview of the performance of our *WizardLM*, we conduct a comparative comparison between our model and the established baselines across a range of LLM benchmarks.

**OpenLLM Leaderboard of HuggingFace** (Beeching et al., 2023) includes MMLU (Hendrycks et al., 2020), ARC (Clark et al., 2018), HellaSwag (Zellers et al., 2019), and TruthfulQA (Lin et al., 2022). MMLU consists of a range of multiple-choice academic questions. ARC is a set of grade-school science questions. HellaSwag is a test of commonsense inference. TruthfulQA measures a model's propensity to reproduce falsehoods. We adopt the evaluate code (Gao et al., 2021) from OpenLLM.

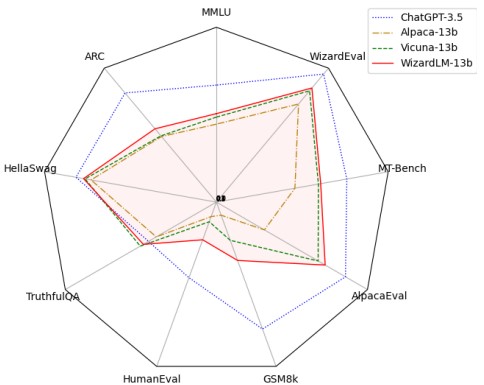

Figure 3: Automatic evaluations on nine LLM benchmarks.

**Code Generation** We use the extensively utilized HumanEval (Chen et al., 2021) benchmark consisting of 164 coding problems to evaluate LLMs' code writing capabilities at the function level by reporting the pass@1 metric.

**Math Reasoning** We use GSM8k (Cobbe et al., 2021) to evaluate mathematical abilities of models, GSM8k contains 1319 grade school math test data. We adopt 4-shot testing and report pass@1.

**GPT-4 Evaluation** We employ two widely recognized GPT-4 evaluation benchmarks, including AlpacaEval (Li et al., 2023c) and MT-Bench (Zheng et al., 2023). We also use GPT-4 to judge LLMs on our following proposed WizardEval.

| Model | Avg. | MMLU | ARC | HellaSwag | TruthfulQA | HumanEval | GSM8k | AlpacaEval | MT-Bench | WizardEval |
|---|---|---|---|---|---|---|---|---|---|---|
| ChatGPT-3.5 | 76.15 | 70.0 | 85.2 | 85.5 | 47.0 | 48.1 | 80.8 | 89.37 | 7.94 | 100.0 |
| Alpaca-13b | 43.44 | 46.63 | 51.20 | 76.31 | 41.62 | 9.2 | 8.35 | 33.25 | 4.78 | 76.6 |
| Vicuna-13b | 54.60 | 50.84 | 51.71 | 79.94 | **52.68** | 12.5 | 24.34 | 70.43 | 6.21 | 86.9 |
| Baize-13b | 51.46 | 49.72 | 56.91 | 79.29 | 47.88 | 14.6 | 8.95 | 66.96 | 5.75 | 81.3 |
| CAMEL-13b | 51.29 | 49.74 | 55.63 | 79.25 | 47.42 | 17.7 | 7.13 | 64.84 | 5.78 | 82.1 |
| Tulu-13b | 52.46 | **53.19** | 53.92 | 80.66 | 43.84 | 21.3 | 36.50 | 45.34 | 5.76 | 79.8 |
| WizardLM-13b | **58.96** | 52.92 | **57.25** | **80.88** | 50.55 | **24.0** | 37.15 | 75.31 | **6.35** | **89.1** |

Table 1: Performance comparison of ChatGPT-3.5, open-source baselines, and WizardLM-13b.

As shown in Figure 3 and Table 1, compared with other same-sized open-sourced models, *WizardLM* has a remarkable performance advantage in most benchmarks. Especially in math, code, and GPT-4 evaluations, it achieves significant improvement over Alpaca, Vicuna, Baize, CAMEL, and Tulu.

---

[5]gpt-3.5-turbo from `https://oai.azure.com/portal`

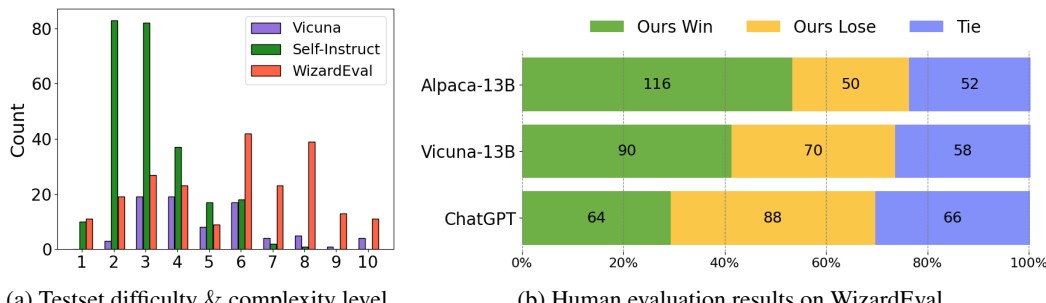

(a) Testset difficulty & complexity level.    (b) Human evaluation results on WizardEval.

Figure 4: WizardEval difficulty and complexity level distribution, and the human evaluation results between WizardLM and baselines (ChatGPT-3.5, Alpaca, Vicuna) on WizardEval.

## 4.4 HUMAN EVALUATION

To evaluate *WizardLM*, we conduct human evaluation on our crafted testbed *WizardEval*, which includes 218 real-world human instructions from diverse sources such as online opensource projects (Github, ShareGPT), platforms (Twitter), and forums (Reddit, Discord). The data contains 29 skills and domains that represent the main requirements of humanity, such as Coding Generation, Math, Reasoning, Complex Formats, Writing, Extensive Disciplines, and so on. As shown in Figure 4a and Appandix Figure 6, we also analyse the difficulty and skills distribution of WizardEval respectively, which indicate that WizadEval is able to handle the evaluation on more complex and demanding scenarios than Self-Instruct and Vicuna testset.

We perform a blind pairwise comparison between *WizardLM*-13b and baselines. Specifically, we recruit 10 well-educated annotators. To each annotator, four responses from Alpaca-13b, Vicuna-13b, *WizardLM* and ChatGPT are presented, which are randomly shuffled to hide their sources. The annotators then judge which response is better following criterion (for detailed definition, please refer to Appendix K): (1) Relevance, (2) Knowledgeable, (3) Reasoning, (4) Calculation, and (5) Accuracy.

Then they should rank the four responses from 1 to 5 (1 means best), and allowing equal scores for comparable instances. To estimate the win rate, we compare the frequency of win, lost, and tie between each pair of models. As shown in Figure 4 (b). *WizardLM* achieved significantly better results than Alpaca and Vicuna, which demonstrates the effectiveness of *Evol-Instruct* method. All of the Kappa scores are greater than 0.6, which indicates the good agreement among the annotators.

## 4.5 ABLATION STUDY

**Training with different data (seed, size), evol model, and base model size.** In order to study the impact of different data seeds, Evol models, scale of evolved dataset, pre-trained models on our proposed method, we conducted the following experiments: a) Using 70k ShareGPT as the seed data to obtain WizardLM-13b (ShareGPT Seed); b) Using LlaMA-2-70B-Chat to replace ChatGPT as the evolutionary execution model to obtain WizardLM-13b (LlaMA-2-70B-Chat Evol); c) We train on larger size pre-trained models Llama-1 65B and Llama-2 70B to obtain WizardLM-65b and WizardLM-70b respectively; d) Using the complete 250k evolved data to obtain WizardLM-13b (250K); e) Using a completely different base from the LlaMA family, Mistral-7B, to obtain WizardLM-7b (Mistral); f) In order to compare more diverse instruction data, we choose Supernatural Instructions(Wang et al., 2022b) and randomly extract 70k data to train llama-13b to obtain LlaMA-13b (SNI). The full results are shown in the Table 2. To investigate the reason of why does WizardLM-13b (ShareGPT Seed) performs worse on GSM8k, we random sample 2000 instructions from ShareGPT and Alpaca data respectively, then use ChatGPT to judge (prompt please refer to Appendix G) whether an instruction is "math" related, we find that the ShareGPT only contains 4.3% math data, and Alpaca data contains 11.8% math data, thus we think that less math data results in worse GSM8k performance of WizardLM-13b (ShareGPT Seed).

The results indicate that (i) the ShareGPT is a better seed for evol-instruct than Alpaca, (ii) larger evolved data size can improve model capacity, and (iii) our proposed Evol-Instruct method is not dependent on ChatGPT, other strong open source model such as Llama-2 is also a good substitute for

| Model | Avg. | MMLU | ARC | HellaSwag | TruthfulQA | HumanEval | GSM8k | AlpacaEval | MT-Bench | WizardEval |
|---|---|---|---|---|---|---|---|---|---|---|
| WizardLM-13b | 58.96 | 52.92 | 57.25 | 80.88 | 50.55 | 24.0 | 37.15 | 75.31 | 6.35 | 89.1 |
| WizardLM-13b (ShareGPT Seed) | 61.87 | 50.92 | 60.24 | 81.39 | 54.56 | 25.0 | 31.46 | 86.32 | 6.76 | 99.3 |
| WizardLM-13b (250K) | 60.30 | 53.78 | 58.53 | 81.39 | 52.26 | 25.6 | 37.46 | 78.10 | 6.51 | 90.3 |
| WizardLM-13b (LlaMA-2-70B-Chat Evol) | 56.27 | 51.09 | 57.34 | 79.12 | 48.76 | 19.5 | 33.83 | 70.47 | 6.18 | 84.5 |
| LlaMA-13b (SNI) | 37.73 | 54.90 | 54.95 | 80.40 | 38.69 | 4.20 | 5.79 | 13.67 | 2.86 | 58.4 |
| Alpaca-7b (Mistral) | 52.87 | 56.34 | 55.38 | 79.49 | 43.92 | 19.2 | 32.05 | 54.26 | 5.47 | 80.5 |
| WizardLM-7b (Mistral) | 65.81 | 60.70 | 57.47 | 82.08 | 51.79 | 37.80 | 59.49 | 80.70 | 7.10 | 91.3 |
| WizardLM-65b | 69.40 | 62.09 | 65.83 | 85.48 | 52.19 | 36.5 | 66.39 | 87.50 | 7.12 | 97.5 |
| WizardLM-70b | 71.33 | 63.32 | 64.52 | 83.21 | 54.60 | 42.1 | 70.61 | 89.32 | 7.46 | 99.7 |

Table 2: WizardLM with different data seed, data size, evol model, and base model size.

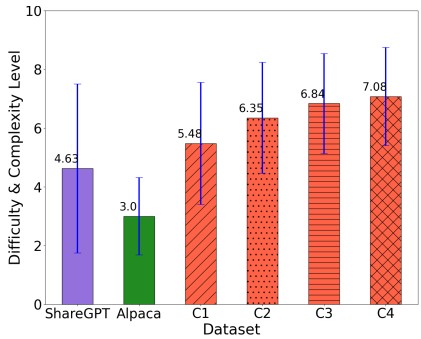

(a) Average difficulty and complexity level

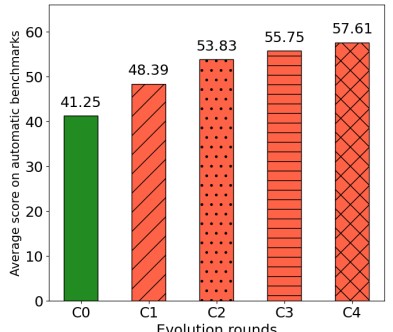

(b) Average score on automatic benchmarks

Figure 5: The difficulty level between ShareGPT, Alpaca, and our four epochs of evolved instruction.

ChatGPT, (iv) our evloved data also shows better finetune performance than Supernatural Instructions. Futhermore, the results on different pre-trained bases (e.g., Llama-1 65B, Llama-2, Mistral-7B) indicate that our Evol-Instruct can be widely applied to various pre-trained models.

**Analysis of In-depth Evolving.** The Figure 5a and 5b presents an ablation study investigating the impact of the number of data evolution rounds. To study the depth of the evolving process, we use ChatGPT to judge the difficulty level of instruction. The used prompt please refer to Appendix E.

Figure 5b shows the average scores (on nine automatic benchmarks in Section 4.3) of the models fine-tuned with the data from each evolution round. Each round of data from $C0$ to $C4$ is about $52k$. From the trend of this figure, it can be seen that as the complexity of the training instruction data gradually increases, the performance of the fine-tuned models also improves synchronously. To investigate the correctness of the difficulty score by ChatGPT, we also use GPT-4 and human to measure the instructions difficulty, the detailed results in the Table 3 of Appendix I indicate the good agreement among the ChatGPT, GPT-4 and human annotators.

**Analysis of In-breadth Evolving.** We aims to examine the semantic breadth of instructions. We use t-SNE van der Maaten & Hinton (2008) and the k-means Hartigan & Wong (1979) algorithm to partition instructions BERT embeddings into 20 clusters. Figure 6 in Appendix F displays clusters, highlighting our method's superior dispersion compared to ShareGPT and Alpaca, indicating greater topic diversity in our instructions.

## 5 CONCLUSIONS

This paper presented Evol-Instruct, an evolutionary algorithm that generates diverse and complex instruction data for LLM. Comprehensive experiments demonstrate that *WizardLM* significantly surpasses typical open-source LLMs such as Alpaca and Vicuna in a wide range of well-recognized benchmarks. Notably, *WizardLM* outperforms baselines by a substantial margin in terms of code, math, GPT-4 and human evaluations.

**Limitations.** This paper acknowledges the limitations of our automatic GPT-4 and human evaluation methods. This method poses challenges for scalability and reliability. Moreover, our test set may not represent all the scenarios or domains where LLM can be applied or compared with other methods.

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

## A    DEEPENING PROMPT

---

**Example A.1: Prompt for Deepening of In-Depth Evolving**

I want you act as a Prompt Rewriter.

Your objective is to rewrite a given prompt into a more complex version to make those famous AI systems (e.g., ChatGPT and GPT4) a bit harder to handle. But the rewritten prompt must be reasonable and must be understood and responded by humans.

Your rewriting cannot omit the non-text parts such as the table and code in #Given Prompt#:. Also, please do not omit the input in #Given Prompt#.

You SHOULD complicate the given prompt using the following method:
**If #Given Prompt# contains inquiries about certain issues, the depth and breadth of the inquiry can be increased.**

You should try your best not to make the #Rewritten Prompt# become verbose, #Rewritten Prompt# can only add 10 to 20 words into #Given Prompt#. '#Given Prompt#', '#Rewritten Prompt#', 'given prompt' and 'rewritten prompt' are not allowed to appear in #Rewritten Prompt#

**#Given Prompt#:**
{Here is instruction.}
**#Rewritten Prompt#:**

---

## B    CONCRETIZING PROMPT

---

**Example B.1: Prompt for Concretizing of In-Depth Evolving**

I want you act as a Prompt Rewriter.

Your objective is to rewrite a given prompt into a more complex version to make those famous AI systems (e.g., ChatGPT and GPT4) a bit harder to handle. But the rewritten prompt must be reasonable and must be understood and responded by humans.

Your rewriting cannot omit the non-text parts such as the table and code in #Given Prompt#:. Also, please do not omit the input in #Given Prompt#.

You SHOULD complicate the given prompt using the following method:
**Please replace general concepts with more specific concepts.**

You should try your best not to make the #Rewritten Prompt# become verbose, #Rewritten Prompt# can only add 10 to 20 words into #Given Prompt#. '#Given Prompt#', '#Rewritten Prompt#', 'given prompt' and 'rewritten prompt' are not allowed to appear in #Rewritten Prompt#

**#Given Prompt#:**
{Here is instruction.}
**#Rewritten Prompt#:**

---

## C    INCREASED REASONING STEPS PROMPT

---

**Example C.1: Prompt for Increased Reasoning Steps of In-Depth Evolving**

I want you act as a Prompt Rewriter.

Your objective is to rewrite a given prompt into a more complex version to make those famous AI systems (e.g., ChatGPT and GPT4) a bit harder to handle. But the rewritten prompt must be

---

reasonable and must be understood and responded by humans.

Your rewriting cannot omit the non-text parts such as the table and code in #Given Prompt#:. Also, please do not omit the input in #Given Prompt#.

You SHOULD complicate the given prompt using the following method:
**If #Given Prompt# can be solved with just a few simple thinking processes, you can rewrite it to explicitly request multiple-step reasoning.**

You should try your best not to make the #Rewritten Prompt# become verbose, #Rewritten Prompt# can only add 10 to 20 words into #Given Prompt#. '#Given Prompt#', '#Rewritten Prompt#', 'given prompt' and 'rewritten prompt' are not allowed to appear in #Rewritten Prompt#

**#Given Prompt#:**
{Here is instruction.}
**#Rewritten Prompt#:**

## D  COMPLICATE INPUT PROMPT

**Example D.1: Prompt for Complicate Input of Evolving**

I want you act as a Prompt Rewriter. Your objective is to rewrite a given prompt into a more complex version using dataformat to make those famous AI systems (e.g., chatgpt and GPT4) more difficult to handle. But the rewritten prompt must be reasonable and must be understood and responded by humans.
You must add [XML data] format text as input data in [Rewritten Prompt]

**#Given Prompt#:**
I'm using this php code to fetch the xml data

**#Rewritten Prompt#:** I have this xml and i want to get the xml data to auto populate HTML table, the codes works but it makes duplicate on the table content

Here is the xml data:

```
<root>
<stats>
<item>
<day>2017-11-01</day>
<impressions>2192</impressions>
<money>1.96790003</money>
</item>
<item>
<day>2017-11-02</day>
<impressions>2824</impressions>
<money>3.208500033</money>
</item>
<item>
<day>2017-11-03</day>
<impressions>3680</impressions>
<money>3.321799981</money>
</item>
</stats>
<total>
<impressions>8696</impressions>
<money>8.498200044</money>
```

```
</total>
<filter>
<dateFrom>2017-11-01</dateFrom>
<dateTo>2017-11-03</dateTo>
<groupBy>day</groupBy>
<format>xml</format>
</filter>
</root>
```

```
 I'm using this php code to fetch the xml data but this code
fetching from whole xml data which makes duplicate field table
```

```php
<?php
\$dom = new DOMDocument;

\$dom -> load('http://example.com/' . \$dateselected . '&dateTo
   =' . \$dateselected2 . '&format=xml');

\$day = \$dom->getElementsByTagName('day');
\\$impressions = \\$dom->getElementsByTagName('impressions');

echo ( "<table>");

    foreach(\\$day as \\$node1) {
    foreach(\\$impressions as \\$node2) {

        echo '<tr>';
        echo "<td>". \\$node1 -> textContent . "<td>";
        echo "<td>". \\$node2 -> textContent . "<td>";
        echo "<td>". \\$node2 -> textContent *0.5/1000 ."<td>";
        echo '</tr>';

    }
}
    echo( "</table>");
?>
```

```
Could anyone give a hint how I can fix this?  thank you

####
```

---

**Example D.2: Prompt for Complicate Input of Evolving**

I want you act as a Prompt Rewriter. Your objective is to rewrite a given prompt into a more complex version using dataformat to make those famous AI systems (e.g., chatgpt and GPT4) more difficult to handle. But the rewritten prompt must be reasonable and must be understood and responded by humans.
You must add [SQL database] format text as input data in [Rewritten Prompt]

**#Given Prompt#:**
achieve the SQL query result

**#Rewritten Prompt#** (MUST contain a specific SQL database as input):
There is a table messages that contains data as shown below:

```
Id Name Other_Columns
-------------------------
1 A A_data_1
2 A A_data_2
3 A A_data_3
4 B B_data_1
5 B B_data_2
6 C C_data_1
```

I If I run a query select * from messages group by name, I will get the result as:

```
1 A A_data_1
4 B B_data_1
6 C C_data_1
```

```
What query will return the following result?
```

```
3 A A_data_3
5 B B_data_2
6 C C_data_1
```

```
That is, the last record in each group should be returned.
At present, this is the query that I use:
```

```
SELECT
  *
FROM (SELECT
  *
FROM messages
ORDER BY id DESC) AS x
GROUP BY name
```

```
But this looks highly inefficient.  Any other ways to achieve
the same result?
```

```
####
```

### Example D.3: Prompt for Complicate Input of Evolving

I want you act as a Prompt Rewriter. Your objective is to rewrite a given prompt into a more complex version using dataformat to make those famous AI systems (e.g., chatgpt and GPT4) more difficult to handle. But the rewritten prompt must be reasonable and must be understood and responded by humans.

You must add [python code] format text as input data in [Rewritten Prompt]
**#Given Prompt#:**
Transformat python code

**#Rewritten Prompt#** (MUST contain a specific python code as input):
I have the following Python code:

```
cursor.execute("INSERT INTO table VALUES var1, var2, var3,")
```

```
where var1 is an integer, var2 and var3 are strings.
How can I write the variable names without Python including them
as part of the query text?
```

```
####
```

---

**Example D.4: Prompt for Complicate Input of Evolving**

I want you act as a Prompt Rewriter. Your objective is to rewrite a given prompt into a more complex version using dataformat to make those famous AI systems (e.g., chatgpt and GPT4) more difficult to handle. But the rewritten prompt must be reasonable and must be understood and responded by humans.
You must add [HTML page] format text as input data in [Rewritten Prompt]

**#Given Prompt#:**
scroll through the whole HTML page

**#Rewritten Prompt#** (MUST contain a specific HTML page as input):
I want to be able to scroll through the whole page, but without the scrollbar being shown.
In Google Chrome it's:

```
::-webkit-scrollbar {
    display: none;
}
```

```
But Mozilla Firefox and Internet Explorer don't seem to work
like that.
I also tried this in CSS:
```

```
overflow: hidden;
```

```
That does hide the scrollbar, but I can't scroll any more.  Is
there a way I can remove the scrollbar while still being able to
scroll the whole page?

With just CSS or HTML, please.
####
```

---

**Example D.5: Prompt for Complicate Input of Evolving**

I want you act as a Prompt Rewriter. Your objective is to rewrite a given prompt into a more complex version using dataformat to make those famous AI systems (e.g., chatgpt and GPT4) more difficult to handle. But the rewritten prompt must be reasonable and must be understood and responded by humans.

You must add [Shell cmd] format text as input data in [Rewritten Prompt]

**#Given Prompt#:**
Shell scp file

**#Rewritten Prompt#** (MUST contain a specific Shell cmd as input):
I'm trying to scp a file from a remote server to my local machine. Only port 80 is accessible.
I tried:

scp -p 80 username@www.myserver.com:/root/file.txt .

but got this error: cp: 80: No such file or directory
How do I specify the port number in a scp command?

####

---

**Example D.6: Prompt for Complicate Input of Evolving**

I want you act as a Prompt Rewriter. Your objective is to rewrite a given prompt into a more complex version using dataformat to make those famous AI systems (e.g., chatgpt and GPT4) more difficult to handle. But the rewritten prompt must be reasonable and must be understood and

---

responded by humans.
You must add [JSON data] format data as input data, add [JSON data] code as input code in [Rewritten Prompt]
Rewrite prompt must be a question style instruction
**#Given Prompt#:**
Given a JSON dataset of customer purchase history, how can we calculate the probability of a customer making a repeat purchase from the same store? Can we utilize the formula for conditional probability: $P(A|B) = P(A \cap B)/P(B)$ where A represents the event of a customer making a repeat purchase and B represents the event of a customer making a purchase from the same store again? Additionally, how can we apply this formula to identify the customer segment that is most likely to make a repeat purchase? Can you provide an example of how to implement this formula using the given JSON dataset?

Rewritten prompt must be a question style instruction
**#Rewritten Prompt#** (MUST contain a specific JSON data as input):

## E   DIFFICULTY JUDGE PROMPT

> **Example E.1: Prompt for Juding the Difficulty of Instructions**
>
> We would like you to evaluate and rate the difficulty and complexity of the following question. You should give an overall score on a scale of 1 to 10, where a higher score indicates higher difficulty and complexity. You must just give a score without any other reasons.
> **## Question:**
> { Here is instruction. }
> **## Score:**

## F   EQUAL PROMPT

> **Example F.1: Prompt for Determining whether Two Instructions are Equal**
>
> Here are two Instructions to ChatGPT AI, do you think they are equal to each other, which meet the following requirements:
> 1. They have same constraints and requirments.
> 2. They have same depth and breadth of the inquiry.
> The First Prompt: {Here is first instruction.}
> The Second Prompt: {Here is second instruction.}
> Your Judgement (Just answer: Equal or Not Equal. No need to explain the reason.):

## G   MATH JUDGEMENT PROMPT

> **Example G.1: Prompt for judging whether an instruction is math related**
>
> Please judge whether the following question is a math problem, and only return True or False without providing any explanation.
>
> Question: {instruction}

## H   WIZARDEVAL ANALYSIS

We collected our *Evol-Instruct* testset that includes real-world human instructions from diverse sources such as online opensource projects, platforms, and forums. We analyzed the data and identified 29 distinct skills that represent the main requirements of humanity, such as Coding Generation & Debugging, Math, Reasoning, Complex Formats, Writing, Extensive Disciplines, and so on. Figure 6 illustrates the distribution of the instances and skills in our test set. Our test set consists of 218

instances, each of which is an instruction for a specific skill. We compared our test set with Vicuna's test set, which is a benchmark dataset for evaluating instruction following models. We found that Vicuna's test set only 80 instances and 9 skills and is much smaller and less diverse than ours. Figure 4a shows how the difficulty and complexity of the test data vary across different instances. Our test data has a more uniform distribution, meaning that it contains instructions with different levels of difficulty and complexity. On the other hand, Vicuna and Alpaca have a skewed distribution, meaning that they mostly contain instructions with low difficulty and complexity. This indicates that these two corpus are not able to handle the evaluation on more complex and demanding scenarios.

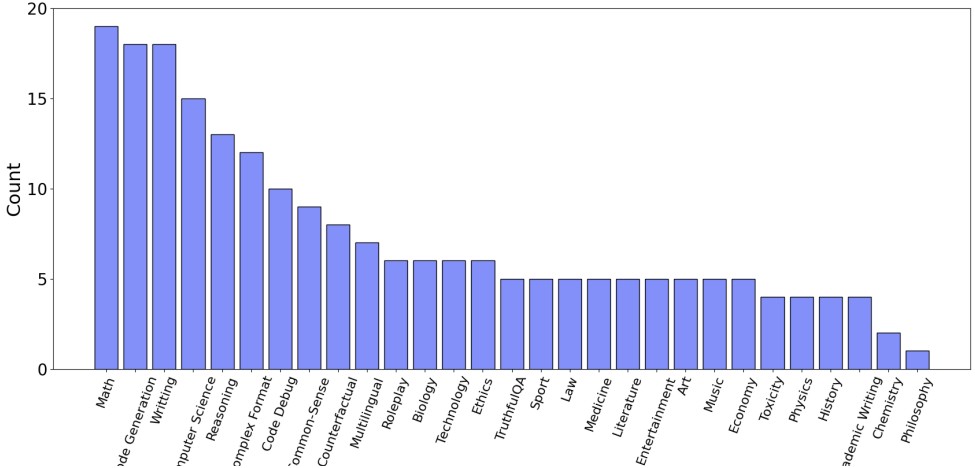

Figure 6: The skills distribution of *Evol-Instruct* testset.

## I  DIFFERENT DIFFICULTY ANNOTATORS

We just use the ChatGPT to post analyse the "difficult" distribution of the generated instructions, but we do not use this analysis results to guide the data generation or model training. In order to explore the ability of ChatGPT to perform difficulty analysis, we sample 600 instructions and use the more powerful GPT4 model and 5 well-educated human annotators together for difficulty assessment. The assessment results are in the Table 3. The results show that ChatGPT, GPT4, and manual annotation show a high degree of consistency in the trend of difficulty changes.

|  | ShareGPT | Alpaca | C1 | C2 | C3 | C4 |
|---|---|---|---|---|---|---|
| GPT-3.5 | 4.63 | 3.00 | 5.48 | 6.35 | 6.84 | 7.08 |
| GPT-4 | 4.31 | 2.69 | 4.68 | 4.90 | 5.37 | 5.54 |
| Human | 4.55 | 3.15 | 5.51 | 5.86 | 6.49 | 6.82 |

Table 3: Use ChatGPT, GPT-4, human to measure the instruction difficulty.

To investigate the correctness of the difficulty score by ChatGPT, we add a new experiment to measure agreement of difficulty judge between ChatGPT and humans: We randomly select two instructions from the six datasets - Alpaca, ShareGPT, C1 to C4 - with equal probability each time, forming a pair. In total, we have selected 300 instruction pairs. Then, we ask ChatGPT and 5 well-educated human annotators to judge which one is more difficulty in one instruction pair, the Kappa score between humans is 0.68, and the Kappa between ChatGPT and human (majority voting) is 0.66, which indicates the good agreement among the ChatGPT and human annotators.

## J  CLUSTER SCATTER PLOT

In-breadth Evolving aims to enhance topic coverage, skill coverage, and overall dataset diversity. To examine (qualitative analysis) the breadth (diversity) of different dataset, we firstly use BERT to

encode each instruction and get its embedding with 768 dimensions, then use a dimension reduction algorithm named t-SNE to reduce embedding dimension to 2, finally we apply a clustering algorithm k-means to partition the instructions of each dataset into 20 clusters for an intuitive visualization. As shown in the Figure 7, the data points of our dataset are more dispersed than ShareGPT and Alpaca (Self-Instruct), which indicates the better topic diversity in our instructions.

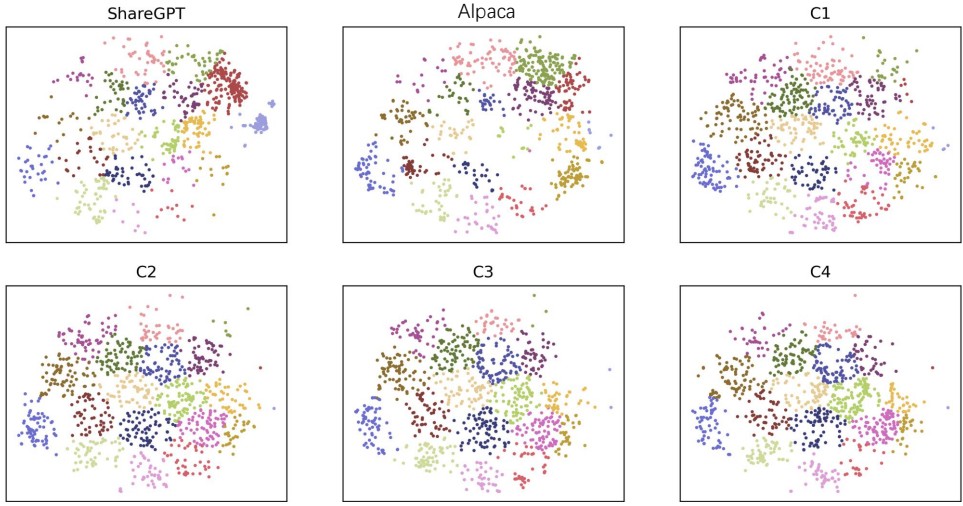

Figure 7: The cluster scatter plot between ShareGPT, Alpaca, and ours four rounds of instruction evolution from C1 to C4. The number of cluster centers is 20.

## K   HUMAN EVALUATION ASPECTS

The annotators then judge which response is better from five aspects:

(1) Relevance: Assessing the model's ability to correctly interpret the semantic meaning of the context and questions.

(2) Knowledgeable: Whether the model can accurately use various and detailed knowledge for problem-solving.

(3) Reasoning: Assessing the model's ability to execute correct reasoning processes or devise valid reasoning concepts to solve problems.

(4) Calculation: Evaluating whether the model can perform accurate mathematical computations of the provided formulas in the domains of math, biology, chemistry and physics.

(5) Accuracy: Evaluating whether the model can perform correctly in the corresponding for a given instruction.

## L   PERFORMANCE DETAILS OF DIFFERENT CHECKPOINTS

In this paper, we train our model with 3 epochs and only reported the performance of the final checkpoint in the above "Section 4 Experiment" to align with previous works.

As shown in the following Table 4, we report the model checkpoints performance on different epochs (2.5, 2,75, 3). For 13B models, we can see that the best performance always appears on WizardLM-13b (ShareGPT Seed) for each benchmark except GSM8k. And for 65b/70b models, we also see that the WizardLM-70b is the best one on all the benchmarks. Therefore, we think this is mainly caused by the fluctuations on some benchmarks in model training.

| Model | Epoch | Avg. | MMLU | ARC | HellaSwag | TruthfulQA | HumanEval | GSM8k | AlpacaEval | MT-Bench | WizardEval |
|---|---|---|---|---|---|---|---|---|---|---|---|
| WizardLM-13b | 2.50 | 57.92 | 52.50 | 56.83 | 78.63 | 49.72 | 22.8 | 35.81 | 74.09 | 6.27 | 88.2 |
| WizardLM-13b | 2.75 | 58.24 | 50.64 | 58.33 | 80.25 | 49.80 | 23.4 | 35.66 | 73.62 | 6.40 | 88.5 |
| WizardLM-13b | 3.0 | 58.96 | 52.92 | 57.25 | 80.88 | 50.55 | 24.0 | **37.15** | 75.31 | 6.35 | 89.1 |
| WizardLM-13b (ShareGPT Seed) | 2.50 | 61.48 | 51.76 | 60.02 | **81.53** | 53.24 | 25.3 | 31.83 | 85.71 | 6.52 | 98.7 |
| WizardLM-13b (ShareGPT Seed) | 2.75 | **62.00** | **53.10** | 58.53 | 79.77 | 54.21 | **27.2** | 33.04 | **86.68** | 6.65 | 99.0 |
| WizardLM-13b (ShareGPT Seed) | 3.0 | 61.87 | 50.92 | **60.24** | 81.39 | **54.56** | 25.0 | 31.46 | 86.32 | **6.76** | **99.3** |
| WizardLM-65b | 2.50 | 68.12 | 60.50 | 63.24 | 84.11 | 50.55 | 35.8 | 66.01 | 86.49 | 7.06 | 95.8 |
| WizardLM-65b | 2.75 | 69.89 | 62.84 | 65.51 | 85.26 | 52.22 | 37.1 | 67.46 | 89.68 | 7.20 | 96.9 |
| WizardLM-65b | 3.0 | 69.40 | 62.09 | 65.83 | 85.48 | 52.19 | 36.5 | 66.39 | 87.50 | 7.12 | 97.5 |
| WizardLM-70b | 2.50 | 71.22 | 61.85 | **66.31** | **85.60** | **54.76** | 41.3 | 68.70 | 87.73 | **7.53** | 99.4 |
| WizardLM-70b | 2.75 | 71.08 | **63.44** | 64.89 | 84.06 | 53.21 | **42.4** | 69.55 | 89.09 | 7.38 | 99.3 |
| WizardLM-70b | 3.0 | **71.33** | 63.32 | 64.52 | 83.21 | 54.60 | 42.1 | **70.61** | **89.32** | 7.46 | **99.7** |

Table 4: Performance details of different checkpoints.

