# OpenReview forum: "WizardLM: Empowering Large Pre-Trained Language Models to Follow Complex Instructions"
_ICLR.cc/2024/Conference — ICLR 2024 poster_

### Official Review · Reviewer_HVLz · 2023-10-26

**Soundness:** 2 fair
**Presentation:** 3 good
**Contribution:** 3 good
**Rating:** 6
**Confidence:** 4

**Summary:**

The paper introduces Evol-Instruct, a data augmentation pipeline to automatically generates diverse and complicated instructions and responses using LLMs. By instruction-tuning a Llama model on the generated instances (the resulting model is referred to as WizardLM), the paper shows that WizardLM outperforms Vicuna and Alpaca on automatic benchmarks and human evaluation.

**Strengths:**

- The proposed method is straightforward and simple.
- The paper conducts both automatic and human-based evaluation, increasing the reliability of the effectiveness of the proposed method.

**Weaknesses:**

- The paper lacks the number of baselines for comparison. For example, there are recent LLaMA-based instruction-tuned models such as Tulu (Wang et al, 2023) and Orca (Mukherjee et al, 2023). However, the paper does not compare the performance nor discuss the related papers.
- The process of generating training instances largely relies on a single LLM (ChatGPT). If the prompts and design choices are optimized towards a single blackbox LLM, the transferability to other LLMs is questionable. Is the proposed data generation method also effective for other LLMs using the same experimental design (Using LLMs other than ChatGPT for training data generation)?
- In Figure 5, the paper calculates the difficulty of the instruction by simply using ChatGPT. However, the instructions are also generated by the same LLM by prompting to make the previous instructions harder. Therefore, the validity of the difficulty estimation used in this paper is questionable. To test whether generated instructions "actually" become more challenging, the paper should calculate the difficulty by using different evaluators or other metrics.

Reference:
Wang et al, 2023: How Far Can Camels Go? Exploring the State of Instruction Tuning on Open Resources
Mukherjee et al, 2023: Orca: Progressive Learning from Complex Explanation Traces of GPT-4

**Questions:**

- For human evaluation, how many labelers labeled the same instances? Could you share the exact inter-labeler agreement between human labelers?
- Does using the full 250K instructions lead to better performance on the automatic evaluation setting?

---

> ### Author Response · Authors · 2023-11-20
> **Response to Reviewer HVLz**
>
> Dear Reviewer, we thank you for your valuable comments and the time you spent reviewing our work!
>
> Please find below a detailed discussion of the points you have raised:
>
> **Weaknesses:**
> > Weakness 1: The paper lacks the number of baselines for comparison.
>
> As shown in the table in the section of **New Baselines** of General Response, we add more baselines such as Baize, CAMEL, Tulu. The extensive results indicate that our WizardLM also significantly surpasses these open source models, which demonstrates that the convincing quality of instruction-following dataset crafted by Evol-Instruct. We don't report the Orca due to it's not an open-source model. We add discussion about Orca in the relate work of updated main submission.
>
> > Weakness 2: The process of generating training instances largely relies on a single LLM (ChatGPT).
>
> To verify that our designed prompt and method can be transferred to other LLMs, we chose LlaMA-2-70B-Chat as a substitute for ChatGPT. Following the same prompts and design choices outlined in the paper, we re-executed the entire evolution process starting from Alpaca data and trained the WizardLM-13b(LlaMA-2-70B-Chat) model with randomly selected 70k data. Please refer to below table for details, although LlaMA-2-70B-Chat's ability is weaker than ChatGPT, WizardLM-13b(LlaMA-2-70B-Chat) still significantly outperforms Alpaca-13b trained solely on Alpaca data and even outperforms Vicuna-13B trained with much more ChatGPT tokens.
>
> Model | Avg. | MMLU | ARC | HellaSwag  | TruthfulQA  |   HumanEval  | GSM8k| AlpacaEval  | MT-Bench  | WizardEval
> | ----- |------| ---- |------|-------| ----- |  ----- | ----- |----- |----- |----- |
> |   Alpaca-13b   | 43.44 | 46.63 | 51.20  | 76.31 |41.62 | 9.2 | 8.35| 33.25 |  4.78  | 76.6
> |   Vicuna-13b   | 54.60  | 50.84 |  51.71 | 79.94 | 52.68  | 12.5  | 24.34 | 70.43  | 6.21 | 86.9
> | WizardLM-13b | 58.96 | 52.92 |  57.25 | 80.88  |50.55 | 24.0 | 37.15 | 75.31  | 6.35  | 89.1
> | WizardLM-13b(LlaMA-2-70B-Chat Evol)  |  56.27|  51.09 | 57.34 | 79.12 | 48.76 | 19.5 | 33.83 | 70.47 | 6.18   | 84.5
>
> > Weakness 3: In Figure 5, the paper calculates the difficulty of the instruction by simply using ChatGPT. The paper should calculate the difficulty by using different evaluators or other metrics.
>
>  In order to explore the ability of ChatGPT to perform difficulty analysis, we sample 600 instructions (each dataset has 100 instructions) and use the more powerful GPT4 model and 5 well-educated human annotators together for difficulty assessment. The assessment results are in the following Table. The results show that ChatGPT, GPT4, and manual annotation show a high degree of consistency in the trend of difficulty changes.
>
> | |   ShareGPT |   Alpaca |   C1 |   C2 |  C3  | C4 |
>  | ----- |------| ---- |------|-------| ----- | ----- |
>  | GPT-3.5 |   4.63 |   3.0 |   5.48 |   6.35 |  6.84  | 7.08 |
>  | GPT-4   |  4.31  |  2.69  |  4.68   |  4.9  |  5.37  |  5.54 |
> | Human   |  4.55  |  3.15  |  5.51   |  5.86  |  6.49  |  6.82 |
>
> **Questions:**
> > Question 1: For human evaluation, how many labelers labeled the same instances? Could you share the exact inter-labeler agreement between human labelers?
>
> As mentioned in the Section 4.4 Human Evaluation, we recruit 10 well-educated annotators to judge the whole test set. All of the Kappa scores are greater than 0.6 (WizardLM vs. Alpaca is 0.67, WizardLM vs. Vicuna is 0.64, WizardLM vs. ChatGPT is 0.63), which indicates the good agreement among the annotators.
>
> > Question 2: Does using the full 250K instructions lead to better performance on the automatic evaluation setting?
>
> Yes, as shown in the following table, we also report the performance of model trained on 250K instructions, it achieves better performance on all the automatic benchmarks than the model trained on 70K instructions.
>
> Model | Avg. | MMLU | ARC | HellaSwag  | TruthfulQA  |   HumanEval  | GSM8k| AlpacaEval  | MT-Bench  | WizardEval
>  | ----- |------| ---- |------|-------| ----- |  ----- | ----- |----- |----- |----- |
>  | WizardLM-13b (70K) | 58.96 | 52.92 |  57.25 | 80.88  |50.55 | 24.0 | 37.15 | 75.31  | 6.35  | 89.1
> | WizardLM-13b (250K) |   60.30 |   53.78 |   58.53 |   81.39 |   52.26 |   25.6 |   37.46 |   78.10 |   6.51 |  90.3 |
>
> \
> We hope these resolve all your concerns about this paper!

---

> > ### Comment · Reviewer_HVLz · 2023-11-22
> > **Questions to Authors**
> >
> > Thank you for your response.
> >
> > I have some further questions and concerns.
> > Weakness 1: The bold should be on Tulu for MMLU, not WizardLM for Table 1. Please update the paper. (53.19 vs 52.92)
> > Weakness 3: How did you define the concept of "difficulty" for the experiments? Since difficulty is inherently subjective (depends on the evaluator), is there any standard to make the boundaries clearer (ex) Provide examples corresponding to the difficulty to the labelers)?
> > Question 1: So does it mean that "each" of the 10 annotators evaluated the "whole" set?
> >
> > Thank you.

---

> ### Author Response · Authors · 2023-11-22
> **Response to the new Questions from Reviewer HVLz**
>
> We would like to thank the reviewer for engaging so thoroughly with both our paper and the rebuttal. We hope the following addresses all remaining concerns:
>
>
> > Weakness 1: The bold should be on Tulu for MMLU, not WizardLM for Table 1. Please update the paper. (53.19 vs 52.92)
>
> Thanks so much for the knidly reminder. We have updated the Table 1 in the latest revision.
>
> > Weakness 3: How did you define the concept of "difficulty" for the experiments? Since difficulty is inherently subjective (depends on the evaluator), is there any standard to make the boundaries clearer (ex) Provide examples corresponding to the difficulty to the labelers)?
>
> Yes. Given an instruction, we let the labelers give an overall score on a scale of 1 to 10, where a higher score indicates a higher difficulty.
>
> In order to clarify and unify the labelers' understanding of difficulty, we have prepared some topics for each with three related instructions on 2, 5, 9 difficulty levels respectively.
>
> Example-1: (General topic)
>
>  Difficulty | Instruction |
> | ----- |------|
>  | 2 | What is global warming?
>   | 5 | Create a question about the consequences and governance of global warming with English, then answer it with French.
>    | 9 | I need detailed data on temperature and carbon dioxide changes in North America over the past decade. Please present it in a markdown format and use Python Matplotlib to plot the trend of both. Finally, based on the phenomena you observe, provide your predictions and suggestions with more than 2000 words.
>
> Example-2: (Code topic)
>  Difficulty | Instruction |
> | ----- |------|
> | 2 | Solve with C++: You are given an array prices where prices[i] is the price of a given stock on the i-th day. You want to maximize your profit by choosing a single day to buy one stock and choosing a different day in the future to sell that stock. Return the maximum profit you can achieve from this transaction. If you cannot achieve any profit, return 0.
> | 5 | Solve with C++: You are given an integer array prices where prices[i] is the price of a given stock on the i-th day. On each day, you may decide to buy and/or sell the stock. You can only hold at most one share of the stock at any time. However, you can buy it then immediately sell it on the same day. Find and return the maximum profit you can achieve.
> | 9 | Solve with C++: You are given an array prices where prices[i] is the price of a given stock on the i-th day. Find the maximum profit you can achieve. You may complete at most two transactions. Note: You may not engage in multiple transactions simultaneously (i.e., you must sell the stock before you buy again).
>
> Example-3: (Math topic)
>  Difficulty | Instruction |
> | ----- |------|
> | 2 | If x - 2 = 0, what is x?
> | 5 | How would you solve for the roots of x^2 - 4x + 4 = 0?
> | 9 | $x = {1+\frac{\sqrt{2}}{1+\frac{\sqrt{2}}{1+...}}}$. Find $\frac{1}{(x+1)(x-2)}$. When your answer is in the form $\frac{A+\sqrt{B}}{C}$, where $A$, $B$, and $C$ are integers, and $B$ is not divisible by the square of a prime, what is $\|A\|+\|B\|+\|C\|$?
>
>
> > Question 1: So does it mean that "each" of the 10 annotators evaluated the "whole" set?
>
> Yes, each annotator has evaluated the whole set.
>
> We want to thank you for you review. Please let us know if any of your points were not addressed properly, or if you have any additional questions.

---

### Official Review · Reviewer_tNq7 · 2023-10-29

**Soundness:** 3 good
**Presentation:** 3 good
**Contribution:** 3 good
**Rating:** 6
**Confidence:** 3

**Summary:**

The paper explores the challenges large language models (LLMs) face in following user-specified instructions. To address these issues, the authors introduce a novel approach, "Evol-Instruct," which uses LLMs to automatically produce open-domain instructions at various complexity levels. By starting with a base set of instructions, the Evol-Instruct method evolves them to increase complexity, either deepening existing instructions or introducing new ones. The resulting instructions are then used to fine-tune the LLaMA model, producing a new model named WizardLM. Through both automatic and human evaluations, WizardLM demonstrated superior performance compared to other models like Alpaca and Vicuna.

**Strengths:**

- Innovative Approach: The Evol-Instruct method offers a fresh perspective on generating complex instructions without relying solely on human input.
- Comprehensive Evaluation: The authors provide both automatic and human evaluations to assess the performance of WizardLM.
- Broad Applicability: WizardLM's superior performance in varied benchmarks, including code, math, and general conversation, suggests its broad applicability.
- Consideration of Instruction Complexity: The paper highlights the significance of instruction complexity in enhancing LLM performance.

**Weaknesses:**

- Reliance on LLMs: The Evol-Instruct method's dependence on LLMs may introduce biases from the original training data.
- Uncertainty in Evolution Direction: The random selection between In-depth and In-breadth evolving may not always yield the optimal instruction complexity.
- Potential Overfitting: The process of evolving instructions multiple times might risk overfitting the model to specific types of instructions.
- Paper Organization: This paper is not well-organized enough, especially for “3 APPROACH”. For example, the “Prompts of In-Depth Evolving” and “Prompts of In-Breadth Evolving” should contained within “Instruction Evolver”, but are listed in a parallel format. There is limited content in “Response Generation”.

**Questions:**

- How do you ensure that the evolved instructions maintain their intended purpose and don't deviate from the original goal?
- Are there any measures to prevent potential biases introduced during the instruction evolution process?
- How does WizardLM handle instructions that are not part of the evolved set?
- Would incorporating more human feedback during the instruction evolution process help in refining the quality of instructions? If so, how?
- How scalable is the Evol-Instruct process for even larger models or datasets in the future?
- What’s your opinion of learning with Evol-Instruct and in-context learning with demonstrations? The correlations and differences, and the pros and cons.

---

> ### Author Response · Authors · 2023-11-20
> **Response to Reviewer tNq7 [1/2]**
>
> Dear Reviewer, we thank you for your valuable comments and the time you spent reviewing our work!
>
> Please find below a detailed discussion of the points you have raised:
>
> **Weaknesses:**
>
> > Weakness 1: Reliance on LLMs.
>
> We agree with your concern about the use of LLMs, thus one of our current research directions is introducing human feedback to the evolution process.
>
> **detailed response:** *We agree with your concern about the use of LLMs. We consider the elimination of potential biases introduced in evolution as an important direction for our research. A feasible method is to use human feedback as a signal to alleviate the inherent bias of the LLM during the evolution process. Although LLM has the issue of bias, the advantage of using it instead of humans for instruction evolution is that the cost is low enough and the speed is faster.*
>
> > Weakness 2: Uncertainty in Evolution Direction:
>
> Yes, this version of Evol-Instruct neglects the detailed control over the direction and topics of evolution, how to make it controllable and in line with specific goals is a topic worth research. For this paper, we mainly want to demonstrate that evolving instructions can improve the supervised fine-tuning performance of pre-trained LLM.
>
> > Weakness 3: Potential Overfitting:
>
> Yes, executing the Evol Instruct multiple times may lead to getting stuck in a specific topic and not being able to cover a wider range of topics. This is also why we propose the branch of In-Breadth Evolving in addition to In-Depth Evolving, which is an attempt to solve this problem. From Chapter 4.5 of the main submission and Appendix F, it can be seen that the diversity of topics in our evolved instruction data is actually better than the original data. How to further improve the problem of overfitting is a topic worth continuous exploration.
>
> > Weakness 4: Paper Organization:
>
> Thanks for your kindly advise, we have reorganized and updated our paper about section "3 APPROACH".
>
> **Questions:**
>
> > Question 1: How do you ensure that the evolved instructions maintain their intended purpose and don't deviate from the original goal?
>
> - (1) To ensure the accuracy of the evolution results, we did a lot of case study and refine the prompts, all the authors have tried almost 300 cases, and the quality meet ours requirements.
> - (2) We also propose a "Elimination Evolving" method, which uses ChatGPT to measure whether the evolved instructions  "have same constraints and requirments" and  have "same depth and breadth of the inquiry" compared to the original one, if they are equal, we will eliminate it.
>
> **detailed response:**
>
> - *(1) To ensure the accuracy of the evolution results, we did a lot of case study and refine the prompts, all the authors have tried almost 300 cases, and the quality meet ours requirements. For example, we find that the Complicating Input of In-Depth Evolving is difficult for ChatGPT to accurately perform evolution in a zero-shot setting. We adopt a few-shot approach, by providing a few appropriate examples, to ensure that this evolutionary method can work properly with ChatGPT.*
> - *(2) We also propose a "Elimination Evolving" method, which uses ChatGPT to measure whether the evolved instructions  "have same constraints and requirments" and  have "same depth and breadth of the inquiry" compared to the original one, if they are equal, we will eliminate it.*
>
> > Question 2: Are there any measures to prevent potential biases introduced during the instruction evolution process?
>
> We fully agree with you that the evolution may introduce some potential biases. We propose two new methods:
> - (1) Use ChatGPT to measure if there is any toxic or bias in the new instruction and answer
> - (2) Use RLHF to improve alignment.
>
> **detailed response:** *We fully agree with you that the evolution may introduce some potential biases. We propose two new methods: (1) Use ChatGPT to measure if there is any toxic or bias in the new instruction and answer (2) Sample some evolved instructions and hire human annotators mark out the possible types of bias in them. Based on this data, train a classifier to filter out those biased evolved instructions during the evolution process.*
>
> > Question 3: How does WizardLM handle instructions that are not part of the evolved set?
>
> To better handle the instruction types as much as possible, we propose the In-Breadth Evolving, which aims to enhance topic coverage, skill coverage, and overall diversity.
>
> **detailed response:** *To better handle the instruction types as much as possible, we propose the In-Breadth Evolving, which aims to evolve a completely brand-new instruction that is different from the current one to be evolved. In-Breadth Evolving can enhance topic coverage, skill coverage, and overall diversity. Section 4.5 of the paper and Appendix F demonstrate that the diversity of topics in our evolved instruction data is actually better than the original data.*

---

> ### Author Response · Authors · 2023-11-20
> **Response to Reviewer tNq7 [2/2]**
>
> > Question 4: Would incorporating more human feedback during the instruction evolution process help in refining the quality of instructions? If so, how?
>
> Sure, human feedback is important in the evolution process, one of our on-going research project is "human-in-the-loop evolution", which indicates that the human feedback would improve the instruction quality and accuracy, we will public related research in the future.
>
> **detailed response:** _Sure, human feedback is important in the evolution process. What we refer to as "human-in-the-loop evolution" aligns very well with your idea of "incorporating more human feedback during the instruction evolution process helps in refining the quality of instructions". This is a direction we are currently exploring. Similar to OpenAI's RLHF method, we evolve an instruction multiple times, hire people to score and rank the evolved instructions. Then, we train a reward model for instruction quality based on this human-annotated data. When executing Evol-Instruct, we select high-scoring instructions based on this reward model and set the weight of the gradient during fine-tuning according to the score. Cases with high instruction scores have a larger gradient update weight, while cases with low scores have a smaller gradient update weight._
>
> > Question 5: How scalable is the Evol-Instruct process for even larger models or datasets in the future?
>
> Besides the WizardLM-13b (52K self-instruct data as the seed), we add the experiments of three new trained wizard models: lager models WizardLM 65b and WizardLM 70b which is trained from Llama-1 65 and Llama-2 70B, and WizardLM-13b (70K ShareGPT data as the seed). The results in the following table indicate that our Evol-Instruct shows scalable performance for larger models or datasets.
>
>
> Model | Avg. | MMLU | ARC | HellaSwag  | TruthfulQA  |   HumanEval  | GSM8k| AlpacaEval  | MT-Bench  | WizardEval
>  | ----- |------| ---- |------|-------| ----- |  ----- | ----- |----- |----- |----- |
>  | WizardLM-13b | 58.96 | 52.92 |  57.25 | 80.88  |50.55 | 24.0 | 37.15 | 75.31  | 6.35  | 89.1
> | WizardLM-13b (ShareGPT Seed) | 61.87 | 50.92 |  60.24 | 81.39 | 54.56 | 25.0 | 31.46 | 86.32 | 6.76 | 99.3
>  | WizardLM-65b   |   69.40 |   62.09 |  65.83 |  85.48 |  52.19 |  36.5 |  66.39 |  87.50 |  7.12 |  97.5 |
>   |  WizardLM-70b   |   71.33 |   63.32 |  64.52 |  83.21 |  54.60 |  42.1 |  70.61 |  89.32 |  7.46 |  99.7 |
>
> > Question 6: What’s your opinion of learning with Evol-Instruct and in-context learning with demonstrations? The correlations and differences, and the pros and cons.
>
> In-context learning can allow Large Language Model (LLM) to learn a new task with only a few examples given. For tasks like Evol-Instruct that make instructions more complex, which are not common in the training data for LLM, enabling in-context learning is expected to help LLM perform better in evolution. In fact, in our In-Depth Evolving of Complicating Input, we use in-context learning to enhance the evolution effect. For details, please refer to Appendix D. In-context learning is a process where the LLM learns a new task based solely on a few given examples, without doing any Back Propagation (BP). Evol-Instruct is aimed at constructing complex instructions to enhance the performance of large pre-trained language model supervised fine-tuning. Therefore, for learning a new task, the advantage of in-context learning is that it requires very few resources, while Evol-Instruct requires more resources but can achieve better results.
>
> We hope these resolve all your concerns about this paper!

---

> ### Comment · Reviewer_tNq7 · 2023-11-22
> **Response to Reviewers based on the First-round Rebuttal**
>
> Dear Authors,
>
> Thank you very much for the response to my proposed issues.
>
> I have reviewed them and most of them can address my concerns. However, some explanation are too simple and lack details. For example,
> * Could you clarify "human-in-the-loop evolution"? How do you make it?
> * The sufficient experimental analysis on the updated results on three new trained wizard models. The analysis in Section 4.5 is a little superficial. For example, Why does WizardLM-13b (ShareGPT Seed) perform worse on MMLU, GSM8k? Why does WizardLM-70b perform worse on ARC and HellaSwag? These observations are contrary to your conclusions in Section 4.5: t (i) the ShareGPT is a better seed for evol-instruct, (ii) larger evolved data size can improve model capacity, and (iii) our proposed Evol-Instruct method is not dependent on ChatGPT, other strong open source model such as Llama-2 is also a good substitute for ChatGPT.
>
> I advise the authors pay more attention to the expression and detail analysis in this paper, and explain my questions more detailedly. I would decrease my scores if I don't receive a revised manuscript (changes had better be highlighted) and circumstantial response with more detailed evidences and analysis.
>
>
> Best Wishes,
>
> Reviewer tNq7

---

> ### Author Response · Authors · 2023-11-22
>
> Dear Reviewer tNq7,
>
> Thank you for your constructive comments and valuable suggestions. We have carefully checked our response to your reviews, and have added some details to **Weakness 1** and **Question 1~4**, with the new complete **detailed responses** highlighted in *italics*. In order to answer your questions about our three new wizards, we have added some experiments. We also have updated the corresponding content into the latest manuscript. We hope the following responses can address your concerns.
>
> > Why does WizardLM-13b (ShareGPT Seed) perform worse on GSM8k?
>
> To investigate the reason, we add a new experiment: we random sample 2000 instructions from ShareGPT and Alpaca data respectively, then use ChatGPT to judge whether an instruction is "math" related, we find that the ShareGPT only contains **4.3\%** math data, and Alpaca data contains **11.8\%** math data, thus we think that less math training data ratio results in worse preformance on GSM8k of WizardLM-13b (ShareGPT Seed).
>
> We have also updated above analysis on "**Section 4.5 ABLATION STUDY - Training with different data (seed, size), evol model, and base model size**" and the ChatGPT judgement prompt on **Appendix G** in the latest manuscript.
>
>
> > Why does WizardLM-13b (ShareGPT Seed) perform worse on MMLU?
> > Why does WizardLM-70b perform worse on ARC and HellaSwag?
>
> As shown in the following two tables, we report the model checkpoints performance on different epochs (2.5, 2,75, 3).
>
> In this paper, we train our model with 3 epochs and only reported the performance of the final checkpoint to align with previous works.
>
> For 13B models, we can see that the best performance always appears on WizardLM-13b (ShareGPT Seed) for each benchmark except GSM8k.
> And for 65b/70b models, we also see that the WizardLM-70b is the best one on all the benchmarks.
>
> Therefore, we think this is mainly caused by the fluctuations on some benchmarks in model training.
>
> We have also updated above analysis and following tables on **Appendix L** in the latest manuscript.
>
> Model | Epoch | Avg. | MMLU | ARC | HellaSwag  | TruthfulQA  |   HumanEval  | GSM8k| AlpacaEval  | MT-Bench  | WizardEval
>  | ----- |------ |------| ---- |------|-------| ----- |  ----- | ----- |----- |----- |----- |
>  | WizardLM-13b                 | 2.50 | 57.92 | 52.50 |  56.83 | 78.63  |49.72  | 22.8 | 35.81 | 74.09 | 6.27 | 88.2 |
>  | WizardLM-13b                 | 2.75 | 58.24 | 50.64 |  58.33 | 80.25  |49.80  | 23.4 | 35.66 | 73.62 | 6.40 | 88.5 |
>  | WizardLM-13b                 | 3.0  | 58.96 | 52.92 |  57.25 | 80.88  |50.55  | 24.0 | **37.15** | 75.31 | 6.35 | 89.1 |
>  | WizardLM-13b (ShareGPT Seed) | 2.50 | 61.48 | 51.76 |  60.02 | **81.53**  | 53.24 | 25.3 | 31.83 | 85.71 | 6.52 | 98.7 |
>  | WizardLM-13b (ShareGPT Seed) | 2.75 | **62.0** | **53.10** |  58.53 | 79.77  | 54.21 | **27.2** | 33.04 | **86.68** | 6.65 | 99.0 |
>  | WizardLM-13b (ShareGPT Seed) | 3.0  | 61.87 | 50.92 |  **60.24** | 81.39  | **54.56** | 25.0 | 31.46 | 86.32 | **6.76** | **99.3** |
>
>
>
> Model | Epoch | Avg. | MMLU | ARC | HellaSwag  | TruthfulQA  |   HumanEval  | GSM8k| AlpacaEval  | MT-Bench  | WizardEval
>  | -----| ----- |------| ---- |------|-------| ----- |  ----- | ----- |----- |----- |----- |
>  | WizardLM-65b             | 2.50    | 68.12 | 60.50 |  63.24 |  84.11 | 50.55 | 35.8 | 66.01 | 86.49 | 7.06 | 95.8 |
>  | WizardLM-65b             | 2.75    | 69.89 | 62.84 |  65.51 |  85.26 | 52.22 | 37.1 | 67.46 | 89.68 | 7.20 | 96.9 |
>  | WizardLM-65b             | 3.0     | 69.40 | 62.09 |  65.83 |  85.48 | 52.19 | 36.5 | 66.39 | 87.50 | 7.12 | 97.5 |
>  | WizardLM-70b             | 2.50   | 71.22 | 61.85 |  **66.31** |  **85.60** | **54.76** | 41.3 | 68.70 | 87.73 | **7.53** | 99.4 |
>  | WizardLM-70b             | 2.75    | 71.08 | **63.44** |  64.89 |  84.06 | 53.21 | **42.4** | 69.55 | 89.09 | 7.38 | 99.3 |
>  | WizardLM-70b             | 3.0     | **71.33** | 63.32 |  64.52 |  83.21 | 54.60 | 42.1 | **70.61** | **89.32** | 7.46 | **99.7** |
>
> We hope you find these revisions address your concerns. Please let us know if any of your points were not addressed properly, or if you have any additional questions.
>
> Best regards,
>
> Paper 6631 Authors.

---

> > ### Comment · Reviewer_tNq7 · 2023-11-22
> > **Response to Reviewers based on the Second-Round Rebuttal**
> >
> > Dear Authors,
> >
> > Thank you for the clarification. Taking into account the updates made in the paper and the discussion in the reviews, I would like to stick to my current score.
> >
> > Best Wishes,
> >
> > Reviewer tNq7

---

> > > ### Author Response · Authors · 2023-11-23
> > >
> > > Dear Reviewer tNq7,
> > >
> > > We genuinely appreciate the time and thoughtful consideration you have dedicated to our work.
> > >
> > > Best regards,
> > >
> > > Paper 6631 Authors.

---

### Official Review · Reviewer_fF8a · 2023-11-01

**Soundness:** 3 good
**Presentation:** 3 good
**Contribution:** 3 good
**Rating:** 6
**Confidence:** 5

**Summary:**

The paper argues that human written instruction tuning data as well as GPT-generated self-instruct data lack difficulty and diversity. To address this issue, they propose a procedure to modify existing self-instruct data of Alpaca and make them more diverse and more difficult. The results show that after finetuning on this dataset, the resulting model outperforms Alpaca and Vicuna on a number of standard academic datasets and a small WizardEval dataset.

----
The authors have satisfactorily resolved my concerns during the rebuttal. Hence, I increased the score from 5 to 6.

**Strengths:**

The paper identifies an important problem: the instruction tuning data lacks diversity and difficulty. I agree this is indeed a problem.

The reported evaluation results are good, and lends support to the claim that the new dataset is better than baselines.

**Weaknesses:**

## Major Issues ##

The authors did not perform a thorough evaluation of the created dataset.
- The authors do not compare against diverse instruction tuning datasets such as Natural Instructions, Supernatural Instructions, and the training data of FLAN-T5.
- The experiments focus on Llama and ignore other LLMs such as T5, Falcon, Mistral (which has a model not instruction tuned), and so on.
- The evaluation of difficulty and diversity by ChatGPT is unconvincing, as the paper presents no evidence that ChatGPT is good at these evaluations. It is also dubious to report only the average score of difficulty and diversity, as these are clearly different concepts.
- The WizardEval dataset is too small and the paper contains little detail about its construction, so we do not know if the dataset can serve as a sound and fair evaluation benchmark.
- I do not understand the argument in the section Analysis of In-breadth Evolving.

One possible evaluation of difficulty and diversity is to divide all the training data into several subsets with increasing difficulty / diversity. If ChatGPT is capable of evaluating the training data, then perhaps more difficult and more diverse training data will lead to better performance. Note that this is different from the current grouping based on evolution iterations, which includes a confounding variable: the number of evolution iterations.

## Minor Issues ##

The prompts to ChatGPT contain a few grammatical issues. For example, "Your objective is to rewrite a given prompt into a more complex version to make those famous AI systems (e.g., ChatGPT and GPT4) a bit harder to handle" should be "a more complex version which those famous AI systems (e.g., ChatGPT and GPT4) find a bit harder to handle". It is the prompt that should be hard to handle, not the AI systems.

"But the rewritten prompt must be reasonable and must be understood and responded by humans." Here, whether the prompt is responded to by humans or by AIs cannot be controlled by the prompt generation network. The authors probably meant that it is possible for humans to respond correctly to the prompt.

However, these are likely minor issues because ChatGPT may not have sufficient logic reasoning to understand the subtle differences.

## Writing ##

The paper makes extensive citations to related papers, including many that are recently posted on arxiv. Though it is good to acknowledge similar work, sometimes the cited papers do not support or relate to the text. For example, in the following sentence：

Human annotators are prone to fatigue and cannot sustain high-intensity work to produce a sufficient
proportion of high-difficulty instructions (Zhang et al., 2023; Xiao et al., 2023; Manakul et al., 2023;
Zhong et al., 2023)

The provided citations are unrelated to whether human annotators can sustain high-intensity work. This may create the impression that the paper attempts to mislead the reader, though it is not the authors' intention to do so.

Elimination evolving is a vague and ungrammatical name. I suggest renaming this step. Perhaps failure handling or incorrect prompt filtering?

**Questions:**

- How do we know the answers to the newly created prompts are correct?
- How do we know the difficulty and diversity scores by ChatGPT are correct? Please do not say because other papers did similar things. The other papers could be wrong, too.

---

> ### Author Response · Authors · 2023-11-20
> **Response to Reviewer fF8a [1/2]**
>
> Dear Reviewer, we thank you for your valuable comments and the time you spent reviewing our work!
>
> Please find below a detailed discussion of the points you have raised:
>
> **Major Issues:**
> >  Weakness 1: The authors do not compare against diverse instruction tuning datasets such as Natural Instructions, Supernatural Instructions, and the training data of FLAN-T5.
>
> For a more comprehensive comparison and study, we extract 70k data from the Supernatural Instructions dataset and fine-tune llama-13b with the same training hyperparameters as ours, resulting in the model LlaMA(SNI). The experimental results, as shown in above Table, indicate that LlaMA(SNI) performs well on classic QA benchmarks, such as MMLU, surpassing Alpaca, Vicuna, and WizardLM. However, its performance is less impressive on math, code and open-domain test-beds like MT-Bench.
>
> Model | Avg. | MMLU | ARC | HellaSwag  | TruthfulQA  |   HumanEval  | GSM8k| AlpacaEval  | MT-Bench  | WizardEval
>  | ----- |------| ---- |------|-------| ----- |  ----- | ----- |----- |----- |----- |
>  | WizardLM-13b | 58.96 | 52.92 |  57.25 | 80.88  |50.55 | 24.0 | 37.15 | 75.31  | 6.35  | 89.1
> | LlaMA(SNI) |  37.73|  54.9|  54.95|  80.4|  38.69|  4.2|  5.79|  13.67 |  2.86 | 58.4
>
>
> > Weakness 2: The experiments focus on Llama and ignore other LLMs such as T5, Falcon, Mistral (which has a model not instruction tuned), and so on.
>
> To verify whether our proposed Evol-Instruct can be applied to pre-train models outside of Llama, we chose to train Mistral-7B on Alpaca data and the evolved Alpaca data respectively. The results of the benchmarks are shown in the following table.
>
> Model | Avg. | MMLU | ARC | HellaSwag  | TruthfulQA  |   HumanEval  | GSM8k| AlpacaEval  | MT-Bench  | WizardEval
>  | ----- |------| ---- |------|-------| ----- |  ----- | ----- |----- |----- |----- |
>  | Alpaca (Mistral) | 52.87 | 56.34 |  55.38 | 79.49  |43.92 | 19.2 | 32.05 | 54.26  | 5.47  | 80.5
> | WizardLM (Mistral) |  65.81 |  60.7|  57.47 |  82.08|  51.79|  37.8|  59.49 |  80.7 |  7.1 | 91.3
>
> > Weakness 3.1: The evaluation of difficulty and diversity by ChatGPT is unconvincing.
>
> We just use the ChatGPT to post analyse the "difficulty" distribution of the generated instructions, but we do not use this analysis results to guide the data generation or model training. In order to explore the ability of ChatGPT to perform difficulty analysis, we sample 600 instructions (each dataset has 100 instructions) and use the more powerful GPT4 model and 5 well-educated human annotators together for difficulty assessment. The assessment results are in the following Table. The results show that ChatGPT, GPT4, and manual annotation show a high degree of consistency in the trend of difficulty changes.
>
> | |   ShareGPT |   Alpaca |   C1 |   C2 |  C3  | C4 |
>  | ----- |------| ---- |------|-------| ----- | ----- |
>  | GPT-3.5 |   4.63 |   3.00 |   5.48 |   6.35 |  6.84  | 7.08 |
>  | GPT-4   |  4.31  |  2.69  |  4.68   |  4.90  |  5.37  |  5.54 |
> | Human   |  4.55  |  3.15  |  5.51   |  5.86  |  6.49  |  6.82 |
>
> To investigate the correctness of the difficulty score by ChatGPT, we add a new experiment to measure agreement of difficulty judge between ChatGPT and humans: We randomly select two instructions from the six datasets - Alpaca, ShareGPT, C1 to C4 - with equal probability each time, forming a pair. In total, we have selected 300 instruction pairs. Then, we ask ChatGPT and 5 well-educated human annotators to judge which one is more difficulty in one instruction pair, the Kappa score between humans is 0.68, and the Kappa between ChatGPT and human (majority voting) is 0.66, which indicates the good agreement among the ChatGPT and human annotators.
>
> > Weakness 3.2: It is also dubious to report only the average score of difficulty and diversity, as these are clearly different concepts.
>
> In the Figure 5, we only shows the average score of "difficulty and complexity" (which are synonyms), and this score do not contains the degree of "diversity".
>  \
> Furthermore for "diversity", as shown in the "Section 4.5 - Analysis of In-breadth Evolving" and Figure 6 in Appendix F , we use t-SNE and k-means over BERT embedding of instructions to indicate the topic diversity between self-intruct, ShareGPT and evol-instruct dataset.

---

> ### Author Response · Authors · 2023-11-20
> **Response to Reviewer fF8a [2/2]**
>
> >  Weakness 4. The WizardEval dataset is too small and the paper contains little detail about its construction, so we do not know if the dataset can serve as a sound and fair evaluation benchmark.
>
> To better introduce our WizardEval, we has updated and added some details to the Appendix on latest version of paper:
>
> - (1) Figure 6 in Appendix G illustrates the distribution of the instances and skills in our WizardEval set. It contains 29 distinct skills that represent the main requirements of humanity, such as Coding Generation & Debugging, Math, Reasoning, Complex Formats, Writing, Extensive Disciplines, and so on. It consists of 218 instances, each of which is an instruction for a specific skill.
>
> - (2) We compared our test set with Vicuna’s test set, which is a benchmark dataset for evaluating instruction following models. We found that Vicuna’s test set only 80 instances and 9 skills and is much smaller and less diverse than ours.
>
> - (3) Figure 4 in main submission shows how the difficulty and complexity of the test data vary across different instances. Our test data has a more uniform distribution, meaning that it contains instructions with different levels of difficulty and complexity. On the other hand, Vicuna and Alpaca have a skewed distribution, meaning that they mostly contain instructions with low difficulty and complexity. This indicates that these two corpus are not able to handle the evaluation on more complex and demanding scenarios.
>
> > Weakness 5:  I do not understand the argument in the section Analysis of In-breadth Evolving.
>
> In-breadth Evolving aims to enhance topic coverage, skill coverage, and overall dataset diversity. To examine (qualitative analysis) the breadth (diversity) of different dataset, we firstly use BERT to encode each instruction and get its embedding with 768 dimensions, then use a dimension reduction algorithm named t-SNE to reduce embedding dimension to 2, finally we apply a clustering algorithm k-means to partition the instructions of each dataset into 20 clusters for an intuitive visualization. As shown in the Figure 6 in the paper, the data points of our dataset are more dispersed than ShareGPT and Alpaca (Self-Instruct), which indicates the better topic diversity in our instructions.
>
> **Minor Issues and Writting**:
>
> Thanks for your kindly suggestions for our grammatical and citations issues. We has updated a new version of paper to improve these aspects, please refer to the new version and thank you again.
>
>
> **Questions:**
>
> > Question 1: How do we know the answers to the newly created prompts are correct?
>
> To improve the alignment between the new prompts and our evolution target, we propose "Elimination Evolving" as shown in the paper,  it use two strategies to judge the quality of instruct: ChatGPT and manual rules.
>
> > Question 2: How do we know the difficulty and diversity scores by ChatGPT are correct? Please do not say because other papers did similar things. The other papers could be wrong, too.
>
> - (1) In this paper, we only use ChatGPT to measure the  "difficulty and complexity" (which are synonyms).  And as mentioned in the response to the Weakness 5, we describe how we measure the diversity score with BERT, t-SNE, and k-means.
>
> - (2) To investigate the correctness of the difficulty score by ChatGPT, we add a new experiment to measure agreement of difficulty judge between ChatGPT and humans:
> We randomly select two instructions from the six datasets - Alpaca, ShareGPT, C1 to C4 - with equal probability each time, forming a pair. In total, we have selected 300 instruction pairs. Then, we ask ChatGPT and 5 well-educated human annotators to judge which one is more difficulty in one instruction pair, the Kappa score between humans is 0.68, and the Kappa between ChatGPT and human (majority voting) is 0.66, which indicates the good agreement among the ChatGPT and human annotators.
>
> We hope these resolve all your concerns about this paper!

---

> > ### Comment · Reviewer_fF8a · 2023-11-22
> >
> > Thank you for the new results, which look very good.
> >
> > However, I cannot find any of these results in the new version of the paper. Why would the authors not want to include these positive results in the paper?
> >
> > I cannot increase my score until the authors have sufficient faith in the results to put them in the paper.
> >
> > The authors also did not do the human study on difficulty as I suggested. If you present the data in groups to the human annotators, the grouping itself could leak some information to the annotators. The interrater agreement using Kappa is a good experiment. 0.66 is not perfect but acceptable.

---

> > > ### Author Response · Authors · 2023-11-22
> > > **Response to Reviewer fF8a**
> > >
> > > Thank you for engaging so thoroughly with both our paper and the rebuttal. Your valuable suggestion strengthen the overall quality of our manuscript.
> > >
> > > > Concern 1: However, I cannot find any of these results in the new version of the paper.
> > >
> > > We added the difficulty analysis results in Appendix H. Due to space limitations, we only added results corresponding to common concerns in the last revision. As you suggested, we have update a new version of our paper. We add the results of Supernatural Instructions and WizardLM (Mistral) in table 2 and the detailed description in 4.5 ABLATION STUDY of updated main submission. Please check to see if all your concerns are resovled in the new paper.
> > >
> > > > Concern 2: The authors also did not do the human study on difficulty as I suggested. If you present the data in groups to the human annotators, the grouping itself could leak some information to the annotators.
> > >
> > > For your suggestion about human study on difficulty, in fact, we didn't fully understand your meaning at the beginning. After reading your latest response, it seems that we have understood a bit. Your concern seems is that our grouping will leak information to the labeler. In all our manual annotation processes, we do not inform the annotator about how we organize the data. We will randomly shuffle the instructions in these six datasets and hide their sources. The annotator only sees the instructions that need to be annotated and the annotation guideline.
> > >
> > > We hope you find these revisions address your concerns. Please let us know if any of your points were not addressed properly, or if you have any additional questions.

---

> > > > ### Comment · Reviewer_fF8a · 2023-11-22
> > > >
> > > > Thank you for the update. I've increased my score from 5 to 6.

---

> > > > > ### Author Response · Authors · 2023-11-22
> > > > >
> > > > > Dear Reviewer fF8a,
> > > > >
> > > > > Thank you for updating your score!
> > > > >
> > > > > Best regards, Paper 6631 Authors.

---

### Official Review · Reviewer_JU8z · 2023-11-05

**Soundness:** 3 good
**Presentation:** 3 good
**Contribution:** 3 good
**Rating:** 6
**Confidence:** 4

**Summary:**

The authors propose a novel pipeline for creating large amounts of instruction data with varying levels of complexity using LLM. The proposed approach starts from a set of seed instructions and first uses Evol-Instruct, a suit of prompt templates that can make LLMs such as ChatGPT to rewrite seed instructions into more complex ones. Then all generated data are used for fine-tuning open-source LLMs such as LLaMA. The authors used the proposed method to collect a set of instruction data and fine-tuned LLaMA into "WizardLM", and conduct a suit of evaluation on various benchmarks. Experimental results show some improvement over representative baselines including Alpaca and Vicuna models.

**Strengths:**

1. The idea of using LLMs to rewrite and synthesize more complex instructions is interesting and intuitive.
2. The paper is overall well-written and easy to follow.
3. The authors evaluate WizardLM on a wide range of datasets/benchmarks and the experimental results look promising.

**Weaknesses:**

1. The technical contribution of the proposed method is not very significant because compared to self-instruct, it is only adding the command for LLM to generate more complex instruction. The success of the proposed method largely depend on the abilities of powerful LLMs such as ChatGPT. While interesting and intuitive, I'm not sure the technical contribution of the manuscript is suitable for conferences such as ICLR.

2. The authors compare WizardLM with Vicuna by using the same amount of generated instructions. However, the cost or token consumption used for collecting the datasets (especially compared with Alpaca) should also be controlled or at least mentioned.

3. The manuscript lacks comparisons with other methods for instruction data generation methods such as Baize, CAMEL, etc. For benchmark results, the role of the seed instructions is very important. I assume the seed data is very different, which could be a major cause of the performance difference. Therefore, more detailed ablation study is required to make the results more convincing.

4. Also, it would be very helpful to test the proposed data generation methods using other LLMs (e.g., LLaMA) instead of ChatGPT to better understand the proposed data synthesis pipeline.

4.

**Questions:**

Please see the above weakness section for questions.

---

> ### Author Response · Authors · 2023-11-20
> **Response to Reviewer JU8z [1/2]**
>
> Dear Reviewer, we thank you for your valuable comments and the time you spent reviewing our work!
>
> Please find below a detailed discussion of the points you have raised:
>
> > Weakness 1.1: The technical contribution of the proposed method is not very significant because compared to self-instruct, it is only adding the command for LLM to generate more complex instruction.
>
> Motivation & Novelty: The Self-Instruct[1] is to address the problem of "human-written instruction data that is often limited in quantity", aiming to "provide an almost annotation-free method" based on LLM to automatically generate more instruction data. Our Evol-Instruct is not aimed at increasing the quantity of instructions but rather explores whether increasing the complexity of individual instruction contributes to improving the supervised fine-tuning performance of LLM. Our novelty lies in: (a) To best of our knowledge, we are the first to explicitly propose that increasing the complexity of instructions is beneficial for improving the instruction-following capabilities of pre-trained language models. (b) We propose a brand-new automated method to enhance the complexity and difficulty of existing instruction data without the need for human annotators.
>
> Both automatic and human evaluations consistently indicate that the model trained using Evol-Instruct (WizardLM) surpasses the baseline model (Alpaca) trained using Self-Instruct with a large margin in a broad spectrum of benchmarks.
>
> \
> reference:\
> [1] Self-Instruct: Aligning Language Models with Self-Generated Instructions
>
> > Weakness 1.2: The success of the proposed method largely depend on the abilities of powerful LLMs such as ChatGPT.
>
> In the experiment, the baselines we compared, Alpaca-13b and Vicuna-13b, their training data also came from ChatGPT. Our experiment shows that under the condition of using the same powerful LLM, the model trained by Evol-Instruct will perform better on a wide range of benchmarks. Further, when we use a weaker model LlaMA-2-70B-Chat to replace ChatGPT to execute Evol-Instruct, the trained model is still stronger than the models (i.e., Alpaca-13b and Vicuna-13b) trained by ChatGPT outputs. Please see the details in our response to your Weakness 4.
>
> > Weakness 2: The authors compare WizardLM with Vicuna by using the same amount of generated instructions. However, the cost or token consumption used for collecting the datasets (especially compared with Alpaca) should also be controlled or at least mentioned.
>
> Our motivation is to investigate whether, under the condition of keeping the number of instructions constant, simply increasing the complexity of each instruction can be beneficial to the SFT performance of LLMs. More token consumption is a reasonable expectation for this approach. We have counted the number of tokens required for generating 70k ShareGPT data, Alpaca data, and our 70k WizardLM data. Detailed statistics are as follows: ShareGPT(70k) 118.3M, Alpaca 3.02M, WizardLM(70k) 18.84M. To verify when the token consumption is exactly the same, our method can still outperforms the self-instruct method for generating Alpaca. We randomly selected 11,300 samples from our 70k data, totaling 3.02M tokens, equivalent to Alpaca. We then trained the Wizard-13b(3M) model. The results, as shown in the following table, indicate that Wizard-13b(3M) still significantly surpasses Alpaca-13b in all benchmarks. It's worth noting that the model we trained with only 3M Tokens can surprisingly perform comparably to the Vicuna-13b trained with 118M Tokens.
>
>
> Model | Avg. | MMLU | ARC | HellaSwag  | TruthfulQA  |   HumanEval  | GSM8k| AlpacaEval  | MT-Bench  | WizardEval
>  | ----- |------| ---- |------|-------| ----- |  ----- | ----- |----- |----- |----- |
>   |   Alpaca-13b   | 43.44 | 46.63 | 51.20  | 76.31 |41.62 | 9.2 | 8.35| 33.25 |  4.78  | 76.6
>   |   Vicuna-13b   | 54.60  | 50.84 |  51.71 | 79.94 | 52.68  | 12.5  | 24.34 | 70.43  | 6.21 | 86.9
>    |  WizardLM-13b (3M)    |  54.92 |  50.73 |  53.48 |  78.51 |  49.06    |  19.3 |  29.96   |  70.17 |  5.93 | 83.8
>    |  WizardLM-13b | 58.96 | 52.92 |  57.25 | 80.88  |50.55 | 24.0 | 37.15 | 75.31  | 6.35  | 89.1
>
> > Weakness 3.1: Comparisons with other methods for instruction data generation methods
>
> As shown in the table in the section of **New Baselines** of General Response, we add more baselines such as Baize, CAMEL, Tulu. The extensive results indicate that our WizardLM also significantly surpasses these open source models, which demonstrates that the convincing quality of instruction-following dataset crafted by Evol-Instruct.

---

> ### Author Response · Authors · 2023-11-20
> **Response to Reviewer JU8z [2/2]**
>
> > Weakness 3.2: More detailed ablation study is required to make the results more convincing.
>
> We agree with you, seed instructions is very important for the model performance. So we choose the training data of Alpaca-13b as the seed instructions for our execution of Evol-Instruct. The main submission's Table 1 experimental results show that, based on the same seed instructions, WizardLM-13b after executing Evol-Instruct significantly outperforms Alpaca-13b on a series of benchmarks with a large margin. Although ShareGPT has different seed instructions from Alpaca data, its quality is widely recognized as far superior to Alpaca data. We chose to compare with Vicuna-13b, which uses ShareGPT as training data, in order to demonstrate that even a relatively low-quality instruction-following dataset, after applying our proposed Evol-Instruct, can surpass higher-quality data. To further verify the effectiveness of our method, we use ShareGPT as a new seed, and apply our evol-instruct method on it to produce a new 70k dataset, then train a model named WizardLM-13b (ShareGPT Seed). As shown in the following table, the average performance across nine benchmarks of this model is even higher than the previous version of WizardLM (the seed is Alpaca data). The consistent results of Alpaca data and ShareGPT as evolutionary seeds indicate that Evol-Instruct can significantly improve the performance of fine-tuning pre-trained language models using specific instruction data.
>
> Model | Avg. | MMLU | ARC | HellaSwag  | TruthfulQA  |   HumanEval  | GSM8k| AlpacaEval  | MT-Bench  | WizardEval
>  | ----- |------| ---- |------|-------| ----- |  ----- | ----- |----- |----- |----- |
>  | WizardLM-13b | 58.96 | 52.92 |  57.25 | 80.88  | 50.55 | 24.0 | 37.15 | 75.31  | 6.35  | 89.1
> | WizardLM-13b (ShareGPT Seed) | 61.87 | 50.92 |  60.24 | 81.39 | 54.56 | 25.0 | 31.46 | 86.32 | 6.76 | 99.3
>
>
> > Weakness 4: Also, it would be very helpful to test the proposed data generation methods using other LLMs (e.g., LLaMA) instead of ChatGPT to better understand the proposed data synthesis pipeline.
>
>
> To verify that our proposed Evol-Instruct method is not dependent on ChatGPT, we chose LlaMA-2-70B-Chat as a substitute for ChatGPT. Following the methods outlined in the paper, we re-executed the entire evolution process starting from Alpaca data and trained the WizardLM(LlaMA-2-70B-Chat) model with randomly selected 70k data. Please refer to below table for details, WizardLM(LlaMA-2-70B-Chat) still significantly outperforms Alpaca-13b trained solely on Alpaca data and even outperforms Vicuna-13B trained with much more ChatGPT tokens.
>
> Model | Avg. | MMLU | ARC | HellaSwag  | TruthfulQA  |   HumanEval  | GSM8k| AlpacaEval  | MT-Bench  | WizardEval
> | ----- |------| ---- |------|-------| ----- |  ----- | ----- |----- |----- |----- |
> |   Alpaca-13b   | 43.44 | 46.63 | 51.20  | 76.31 |41.62 | 9.2 | 8.35| 33.25 |  4.78  | 76.6
> |   Vicuna-13b   | 54.60  | 50.84 |  51.71 | 79.94 | 52.68  | 12.5  | 24.34 | 70.43  | 6.21 | 86.9
> | WizardLM-13b | 58.96 | 52.92 |  57.25 | 80.88  |50.55 | 24.0 | 37.15 | 75.31  | 6.35  | 89.1
> | WizardLM-13b(LlaMA-2-70B-Chat Evol)  |  56.27|  51.09 | 57.34 | 79.12 | 48.76 | 19.5 | 33.83 | 70.47 | 6.18   | 84.5
>
> \
> We hope these resolve all your concerns about this paper!

---

> > ### Comment · Reviewer_JU8z · 2023-12-05
> > **Response to Authors Response**
> >
> > Thanks for the authors' response and additional experiments. The results partially resolves my concerns and make the empirical results more solid. I have raised my score from 5 to 6.

---

### Author Response · Authors · 2023-11-20
**General Response**

We thank all reviewers for their insightful and constructive comments, we have made the following revisions to present a more comprehensive work:

**Revisions of Submission and Appendix:**

1. We add more baselines such as Baize, CAMEL and Tulu for comparisons with other methods for instruction data generation methods. The extensive results indicate that our WizardLM also significantly surpasses these open source models, which demonstrates that the convincing quality of instruction-following dataset crafted by Evol-Instruct  (Table 1) .

2. We have rewritten and updated the Approach, Experiment, Appendix to clarify the methods and the experiments details of this paper.

3. We add the ablation experiments about training WizardLM with different data (seed, size), evol method, and base model size (Section 4.5).

4. We expanded and organized the explanation of instruction difficulty score and instruction diversity  to take a further step to understand its intrinsic mechanism (Section 4.5).

5. Besides, to make room for the above experiments, some of the ablations (on Different difficulty Annotators and WizardEval testset.) are now adjusted to Appendix G and H.

\
**New Baselines:**

Model | Avg. | MMLU | ARC | HellaSwag  | TruthfulQA  |   HumanEval  | GSM8k| AlpacaEval  | MT-Bench  | WizardEval
 | ----- |------| ---- |------|-------| ----- |  ----- | ----- |----- |----- |----- |
  |   ChatGPT-3.5  | 76.15 | 70.0 |  85.2  | 85.5  |  47.0  | 48.1  | 80.8 | 89.37  | 7.94 | 100.0
  |   Alpaca-13b   | 43.44 | 46.63 | 51.20  | 76.31 |41.62 | 9.2 | 8.35| 33.25 |  4.78  | 76.6
  |   Vicuna-13b   | 54.60  | 50.84 |  51.71 | 79.94 | 52.68  | 12.5  | 24.34 | 70.43  | 6.21 | 86.9
   |  Baize-13b    |  51.46 |  49.72 |  56.91 |  79.29 |  47.88    |  14.6 |  8.95   |  66.96 |  5.75 | 81.3
   |  CAMEL-13b |  51.29 |  49.74|  55.63|  79.25|  47.42|  17.7 |  7.13|  64.84| 5.78 | 82.1
   |  Tulu-13b |  52.46 |  53.19|  53.92 |  80.66 |  43.84 |  21.3 |  36.5 | 45.34 |  5.76 | 79.8
   |  WizardLM-13b | 58.96 | 52.92 |  57.25 | 80.88  |50.55 | 24.0 | 37.15 | 75.31  | 6.35  | 89.1

\
All code to reproduce the above experiments will also be included in the final open-sourced repo.

In closing, we thank the Reviewers again for their time and valuable feedback. If there are further concerns, please let us know.

---

### Meta-Review · Area_Chair_p8H5 · 2023-12-05

**Metareview:**

This paper introduces a new approach to generate instruction tuning data for LLMs. The key idea is to prompt a strong teacher LLM to “evolve” existing instructions generated by the self-instruct approach in order to make those instructions more complex and diverse. Experiments show that Llama and Mistral models fine-tuned on the generated instruction-following data outperforms models trained on similar amounts of self-instruct data. Ablations also show that the proposed approach is relatively robust against the choice of teacher LLMs and seed instructions.

**Strengths:**

* This paper tackles an important problem of synthetic generation of instruction tuning data to save laborious labeling cost (fF8a). The paper also highlights the importance of modeling and improving instruction complexity (tNq7), which is an intuitive idea for synthetic instruction generation (JU8z, HVLz)

* Nice empirical results against self-instruct and training using user-issued instructions on shareGPT (JU8z, fF8a). All reviewers agree that the extensive automatic and human-based evaluations also make the results more sound.

**Resolved Weaknesses:**

Most issues in the initial version of the draft were resolved in the discussion phase:

* The approach was limited to using ChatGPT as the teacher model and the Alpaca data as the seed instructions (JU8z, HVLz). This was resolved with additional ablations using Llama-chat as the teacher LLM and shareGPT data as seed instructions.

* Missing comparison with other instruction following LLMs (mostly Llama derivatives) or instruction tuning datasets (JU8z, fF8a, HVLz), which was resolved after the authors included more baselines.

* Concerns around the metric to measure prompt complexity by prompting the same teacher LLM chatGPT (fF8a, HVLz), which was resolved with an analysis of the agreement between GPT ratings of prompt complexity and human judgment (reasonable Kappa score).
Many issues with writing and paper organization (tNq7, fF8a)

**Open Issues:**

After several revisions, there are still many issues with the current version in terms of writing and the some analysis of experimental results:

Writing Issues:

* “Elimination Evolving”  (Sec 3.2) is a vague and ungrammatical term (fF8a). Consider changing to “pruning (or filtering) instructions”.
Some citations in the paper are misleading and do not support the claims in their associated text. Some spurious citations spotted by fF8a were fixed, but it’s unclear whether there are still such improper citations. Please carefully review all the citations in the paper to make sure they actually support your claims.

* Citation entries that are not part of a sentence should be put inside parentheses (OpenAI trained GPT-3 **(Brown et al. (2020))**).

* Other writing issues spotted by AC:
  * “closed domain instruction fine-tune” -> “closed-domain instruction tuning”, similarly for “open domain …” in the paragraph below.
Sec 3.3: move the full prompt (without …) to Appendix, and replace the prompt prefix here with a high-level, intuitive summary of the entire prompt.
  * The first bullet point under “Elimination Evolving” is hard to understand: how do you define “information gain” here? It’s not clear how the delta of information is compared between two instructions x, y, especially when y is an “in-breadth” example of x with entirely different topics. Please add more explanations here and also provide examples of accepted/rejected instructions in this step.
Inconsistent usage of “Figure”, “Fig.”, “Section”, “Sec.”.
  * The first `\paragraph{}` of Section 4.5 packed many different ablation results and claims altogether, making it very difficult to understand. Please split those results into separate `\paragraph{}`s, such as “impact of seed instructions”, “impact of teacher models”, “impact of base models”, “impact of instruction size”, “scaling analysis”, “other ablations”.

Issues with the analysis of some experiments:

* The analysis of “in-breadth evolving” method in Sec 4.5 is solely based on the visual shape of t-SNE visualization, which is too vague and “hand-wavy” (fF8a). AC: The authors may consider presenting more quantitative analysis, such as showing the distribution of instructions by topics similarly to arxiv.org/pdf/2308.07124.pdf (Figure 2).

I strongly suggest the authors fix the above issues with writing and analysis, and carefully check the grammar of the entire draft (including prompts, fF8a). Some prompt examples in the main text can be moved to the appendix to accommodate extra contents.

Despite those issues, all the reviewers are leaning towards a positive view of this paper considering the significance of the problem, the nice empirical results and extensive ablations/evaluation. We also believe that the identified issues should be effectively addressed in the final camera-ready version of the paper. Therefore, the recommendation is to accept the submission.

**Justification For Why Not Higher Score:**

While the annotation-free instruction tuning problem considered in this paper is a timely, important and exciting direction, given the issues listed above and overall review scores, accepting this paper as a poster presentation would be more reasonable.

**Justification For Why Not Lower Score:**

Despite those issues, all the reviewers are leaning towards a positive view of this paper considering the significance of the problem, the nice empirical results and extensive ablations/evaluation. The identified issues regarding writing and technical presentations could be effectively addressed in the final camera-ready version of the paper.

---

### Decision · Program_Chairs · 2024-01-16

Accept (poster)